# CONTROLLABLE ADVERSARIAL MAKEUP FOR PRIVACY VIA TEXT-GUIDED DIFFUSION

## ABSTRACT

As face recognition becomes more widespread in government and commercial services, its potential misuse raises serious concerns about privacy and civil rights. To counteract this threat, various anti-facial recognition techniques have been proposed, which protect privacy by adversarially perturbing face images. Among these, generative makeup-based approaches are the most widely studied. However, these methods, designed primarily to impersonate specific target identities, can only achieve weak dodging success rates while increasing the risk of targeted abuse. In addition, they often introduce global visual artifacts or a lack of adaptability to accommodate diverse makeup prompts, compromising user satisfaction. To address the above limitations, we develop `MASQUE`, a novel diffusion-based framework that generates localized adversarial makeups guided by user-defined text prompts. Built upon precise null-text inversion, customized cross-attention fusion with masking, and a pairwise adversarial guidance mechanism using images of the same individual, `MASQUE` achieves robust dodging performance without requiring any external identity. Comprehensive evaluations on open-source facial recognition models and commercial APIs demonstrate that `MASQUE` significantly improves dodging success rates over all baselines, along with higher perceptual fidelity preservation, stronger adaptability to various makeup prompts, and robustness to image transformations.

## 1 INTRODUCTION

Facial recognition (FR) systems have been adopted in a wide range of security, biometrics, and commercial applications (Parkhi et al., 2015). However, their unregulated deployment poses privacy risks, allowing unauthorized surveillance and malicious tracking. To address these concerns, anti-facial recognition (AFR) technologies have emerged to protect user privacy from unauthorized FR systems (Wenger et al., 2023). AFR techniques vary depending on which stage of facial recognition they disrupt and typically work by modifying images before they are shared online. Among them, adversarial methods are particularly effective, subtly altering face images to evade detection while preserving their natural appearance. Traditional adversarial approaches, such as noise-based methods (Joon Oh et al., 2017; Yang et al., 2021; Zhong & Deng, 2022), obscure facial features with norm-bounded global perturbations, while patch-based techniques (Xiao et al., 2021; Komkov & Petiushko, 2021) optimize adversarial patterns in localized image regions. Unfortunately, these methods often introduce noticeable visual artifacts, compromising the usability of the produced face images.

To overcome these limitations, recent work has shifted toward generative approaches, leveraging generative adversarial networks (GANs) or diffusion models for anti-facial recognition. Instead of relying on pixel-wise perturbations, these methods generate unrestricted yet semantically meaningful modifications that blend naturally into facial features. A notable direction is the makeup-based AFR (Yin et al., 2021; Hu et al., 2022; Shamshad et al., 2023; Sun et al., 2024; Fan et al., 2025), which seamlessly integrates adversarial perturbations into makeup—a plausible approach, as makeup is inherently associated with facial appearance (see Section 2 for detailed discussions of related work).

While makeup-based AFR techniques offer a promising balance between privacy protection and aesthetics, they often struggle to preserve fine-grained facial details or fully adhere to user instructions from diverse prompts. In addition, these methods require images of an external target identity to guide the generation process for adversarial makeup transfer and primarily focus on the impersonation

Table 1: Summary of the key distinctive features of `MASQUE` compared with prior AFR methods.

| | Dodging | External ID | Image Quality | Localization | Guidance | Prompt Following |
|---|---|---|---|---|---|---|
| TIP-IM (Yang et al., 2021) | ✗ | ✓ | Low | ✗ | - | - |
| AMT-GAN (Hu et al., 2022) | ✗ | ✓ | Low | ✗ | Image | Low |
| C2P (Shamshad et al., 2023) | ✓ | ✓ | Medium | ✗ | Text | Medium |
| DiffAM (Sun et al., 2024) | ✗ | ✓ | High | ✓ | Image | Medium |
| FPP (Salar et al., 2025) | ✗ | ✓ | High | ✗ | - | - |
| MASQUE (ours) | ✓ | ✗ | High | ✓ | Text | High |

setting for evaluation, increasing the risks of targeted abuse. When considering more privacy-critical dodging scenarios, their performance in protection success rates often drops significantly.

**Contributions.** Recognizing a few limitations of prior state-of-the-art AFR methods (Section 4.1), we develop `MASQUE`, a novel diffusion-based image editing framework for localized adversarial makeup generation with customized text guidance (Section 4.2). In particular, `MASQUE` stands out by achieving the following desiderata simultaneously (see Table 1 for a summary of its key distinctive features):

- *Inoffensive Identity Protection.* Built upon pairwise adversarial guidance, `MASQUE` ensures high dodging success rates without requiring face images of any external identity other than the victim's. Such a property reduces the risks of targeted misuse and ethical concerns.
- *Localized Modification.* `MASQUE` employs a facial mask generation and regularization module to constrain adversarial perturbations to designated areas, while preserving fine-grained details of the original face image, thereby vastly enhancing the visual quality of protected images.
- *Strong User Control.* By utilizing cross-attention fusion with masking, `MASQUE` achieves strong prompt-following capability and is adaptable to diverse makeup prompts, providing greater user control and convenience than existing state-of-the-art AFR methods.

Through comprehensive experiments across FR models and makeup text prompts, `MASQUE` not only achieves significantly higher dodging success rates over baseline methods, but is also capable of preserving the visual quality and fine details of the original face images (Section 5). We also show that `MASQUE` is robust to more diverse makeup prompts and various image transformations (Section 6). All of the above suggest the potential of `MASQUE` as a superior solution to real-world AFR applications.

## 2 RELATED WORK

**Anti-Facial Recognition.** Earlier works adopted obfuscation techniques to obscure the facial identity features (Newton et al., 2005) or craft $\ell_p$ perturbations to fool FR models (Yang et al., 2021). While effective, these methods often compromise image quality, limiting their real-world applicability. Poisoning-based approaches (Cherepanova et al., 2021; Shan et al., 2020) introduce a new protection scheme by injecting subtle adversarial noise into images to degrade the effectiveness of recognition models. These techniques excel at disrupting model training without visibly altering the image but rely on strong assumptions about when and how the unauthorized FR model is constructed. Recently, adversarial makeup (Yin et al., 2021; Hu et al., 2022; Shamshad et al., 2023; Sun et al., 2024; Fan et al., 2025) has emerged as a promising solution to realize the goal of facial privacy protection. These methods leverage the strong generative capability of generative models to embed adversarial perturbations into natural makeup-based facial modifications, deceiving attackers' facial recognition models while preserving the aesthetic appeal. For example, Hu et al. (2022) proposed AMT-GAN, which introduces a regularization module and a joint training pipeline for adversarial makeup transfer within the GAN framework (Goodfellow et al., 2014). Advancements in generative models have enhanced the performance of adversarial makeup transfer, such as Clip2Protect (C2P) (Shamshad et al., 2023) and DiffAM (Sun et al., 2024), which adopt a StyleGAN model or a diffusion-based framework, to enable seamless adversarial face modifications with much improved visual quality. In this work, we build upon these developments by leveraging diffusion models to generate visually consistent, localized adversarial makeup for facial privacy protection under dodging scenarios.

More recently, a line of research (Liu et al., 2023; 2024; Wang et al., 2025; Han et al., 2025; Salar et al., 2025) proposed leveraging diffusion models to generate imperceptible AFR-protections without focusing on makeup. For instance, DiffProtect (Liu et al., 2023) uses the diffusion autoencoder model

and adversarially guides the semantic code of the original image towards a target face, whereas Salar et al. (2025) improved the prior approach by learning unconditional embeddings as null-text guidance and adversarially modifying the latent code in the latent diffusion model. Our work complements these methods, where we focus on improving the state-of-the-art makeup-based AFR approaches without relying on images from any external identity, while preserving the visual quality of original faces with a strong emphasis on localizing and controlling the added perturbations through makeup.

**Diffusion-Based Image Editing using Text Guidance.** Text-guided diffusion models, capable of synthesizing high-quality images from natural language descriptions, significantly promote the popularization of generative AI. Built on denoising diffusion probabilistic models (Ho et al., 2020), they iteratively refine random noise into coherent images guided by text prompts. Recent advances in diffusion models have expanded their capabilities to tasks such as localized editing and controllable generation. Mask-based approaches (Couairon et al., 2022; Avrahami et al., 2023) achieve local text-guided modifications by incorporating user-defined constraints, like masks, to confine edits to specific regions. While effective in preserving unedited areas, these methods often struggle to maintain structural consistency within the edited regions, especially in complex scenarios. On the other hand, mask-free attention-based methods (Hertz et al., 2022; Cao et al., 2023; Tumanyan et al., 2023) use attention injection mechanisms to guide edits without requiring explicit masks. These methods excel at preserving the global structure of the image but are prone to editing leakage, where changes unintentionally affect areas beyond the intended region. In this work, we leverage mask-based cross-attention guidance (Mao et al., 2023) to achieve localized adversarial perturbations in the form of makeup, ensuring precise modifications to desired regions, guided by user-specified prompts, while addressing the limitations of structural inconsistencies seen in previous methods.

# 3 BACKGROUND AND PROBLEM FORMULATION

## 3.1 FACIAL RECOGNITION

We consider an adversarial problem setup, where an attacker adopts unauthorized facial recognition models to identify benign users from their publicly shared facial images. This enables the extraction of sensitive information, posing significant privacy risks. Since many well-trained FR models are readily available—either as open-source implementations or through commercial APIs—attackers can easily obtain automated FR tools to achieve this malicious objective. Formally, let $\mathcal{X} \subseteq \mathbb{R}^n$ be the input space of face images and $\mathcal{Y}$ be the set of possible identities. Given a collection of online-scraped face images $\mathcal{D}$ (corresponding to multiple identities), the adversary aims to correctly identify victim users from their facial images as many as possible: $\max \sum_{(\boldsymbol{x},y)\in\mathcal{D}} \mathbb{1}(\mathrm{FR}(\boldsymbol{x}) = y)$, where $\boldsymbol{x}$ stands for a face image, $y$ is the corresponding ground-truth identity, and $\mathrm{FR} : \mathcal{X} \to \mathcal{Y}$ denotes a FR model. A typical FR model consists of a feature extractor, a gallery database, and a query matching scheme (see Wenger et al. (2023) for detailed descriptions). Since FR models can vary in design, the attacker may utilize different feature extractors and query-matching schemes. To comprehensively assess performance under black-box conditions, we evaluate a range of well-trained FR models, from open-source models to commercial APIs. In addition, we consider face verification as a specialized FR variant, where the gallery database contains only a single user's images.

## 3.2 ANTI-FACIAL RECOGNITION

The goal of AFR is to evade face recognition, thereby improving the protection of benign users' privacy. We focus on the most popular adversarial-based AFR techniques, which craft imperceptible or naturalistic perturbations (e.g., adversarial makeup) to users' face images to fool FR models. Specifically, the objective can be cast into a constrained optimization problem detailed as follows:

$$\max \frac{1}{|\mathcal{D}|} \sum_{(\boldsymbol{x},y)\in\mathcal{D}} \mathbb{1}\big\{\mathrm{FR}\big(\mathrm{AFR}(\boldsymbol{x})\big) \neq y\big\}, \quad \text{s.t.} \quad \frac{1}{|\mathcal{D}|} \sum_{(\boldsymbol{x},y)\in\mathcal{D}} \Delta\big(\mathrm{AFR}(\boldsymbol{x}), \boldsymbol{x}\big) \leq \gamma, \qquad (1)$$

where $\mathrm{AFR} : \mathcal{X} \to \mathcal{X}$ denotes the AFR perturbation function, $\Delta$ stands for a metric that measures the visual distortion of the AFR-perturbed image $\mathrm{AFR}(\boldsymbol{x})$ with reference to the original image $\boldsymbol{x}$, and $\gamma > 0$ is a threshold parameter reflecting the distortion upper bound that can be tolerated.

As characterized by the optimization objective in Equation 1, a desirable AFR technique is expected to achieve a high *dodging success rate* (DSR) against facial recognition, which is the primary evaluation

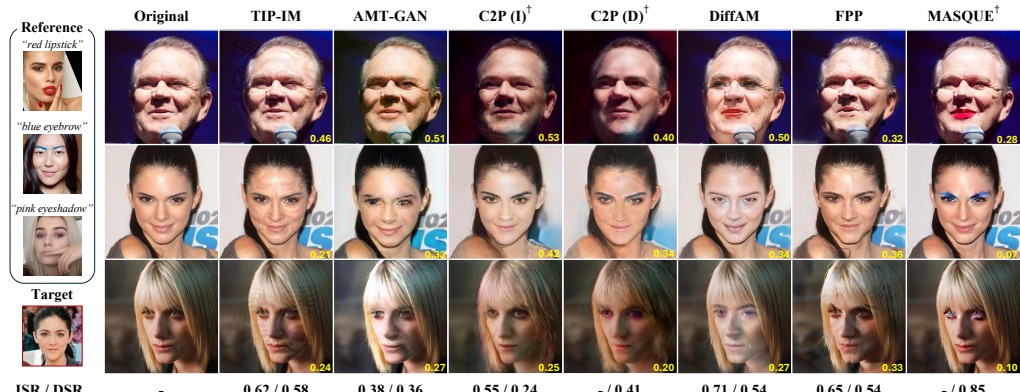

Figure 1: Each row presents the original image followed by AFR-protected images. Reference face images with makeup, text makeup prompts, and the external identity's image are shown under the first column. Methods marked with † require only a text prompt for makeup application. The yellow text indicates the cosine similarity score. Below the images, we report the averaged ISR and DSR.

metric when comparing the effectiveness of different AFR techniques. For simplicity, we also refer to these users as defenders, who apply AFR techniques to modify their facial images with adversarial perturbations. Since the defender typically lacks precise knowledge of the FR model employed by the attacker, we consider the black-box scenario when evaluating the DSR of each AFR method.

From the defender's perspective, images perturbed by AFR should appear natural and closely resemble the original, as captured by the optimization constraint in Equation 1. If $\text{AFR}(\boldsymbol{x})$ has an unsatisfactory image quality, users may be reluctant to share the protected face image online, even if a high DSR is achieved. Given this expectation of high visual quality, earlier privacy-preserving techniques such as face obfuscation or anonymization (Newton et al., 2005; Sun et al., 2018) are not suitable. In comparison, makeup-based AFR methods (Shamshad et al., 2023; Sun et al., 2024) stand out, as they introduce natural adversarial perturbations with minimal visual distortion. In this work, we aim to improve the performance of generative makeup-based AFR, enhancing both DSR and visual quality.

## 4 OUR METHODOLOGY

### 4.1 LIMITATIONS OF PRIOR WORK

First, we conduct preliminary experiments to better understand prior AFR methods in terms of privacy protection and visual quality. Figure 1 shows qualitative comparisons of three makeup styles ("red lipstick", "blue eyebrow", and "pink eyeshadow"), evaluating several adversarial-based AFR methods, including one of the best current noise-based methods, TIP-IM (Yang et al., 2021), three state-of-the-art makeup-based approaches (Hu et al., 2022; Shamshad et al., 2023; Sun et al., 2024), and most recent non-makeup diffusion-based approach (Salar et al., 2025). For each method, we report both the metrics of impersonation and dodging success rates across four black-box facial verification models averaged across 100 randomly selected images from CelebA-HQ (Karras et al., 2017) and the three makeup styles. To ensure a fair comparison with image-guided makeup-based AFR methods, we either select the corresponding makeup image from a benchmark facial makeup dataset (Li et al., 2018) or generate one by inpainting the makeup on a non-makeup image (see Section 5.1 and Appendix B for more experimental details of this preliminary study).

**Weak Protection under Dodging.** In contrast to the untargeted dodging objective, most prior work on anti-facial recognition (Yang et al., 2021; Hu et al., 2022; Shamshad et al., 2023; Sun et al., 2024; Salar et al., 2025) has focused on targeted impersonation, where a user's facial image is adversarially modified to be recognized as a specific target identity $y_t$ ($y_t \neq y$). These methods are typically evaluated by the *impersonation success rate* (ISR), which is defined as follows:

$$\text{ISR} := \frac{1}{|\mathcal{D}|} \sum_{(\boldsymbol{x}, y) \in \mathcal{D}} \mathbb{1}\big\{\text{FR}\big(\text{AFR}_\text{I}(\boldsymbol{x})\big) = y_t\big\}. \tag{2}$$

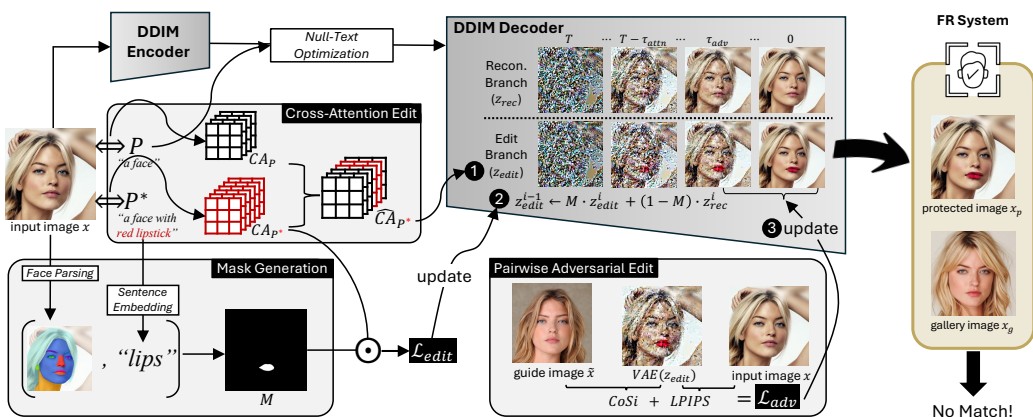

Figure 2: The pipeline of **MASQUE** involves: (1) fusing the editing and reconstruction prompts to produce an updated cross-attention map for diffusion, (2) creating a mask $\mathcal{M}$ to define a target region and optimize an edit loss to maximize makeup-related attention in $\mathcal{M}$, and (3) using pairwise adversarial guidance with same-identity image to enhance identity protection without external targets.

However, ISR and the DSR reflect different goals. As Zhou et al. (2024) shows, high ISR does not necessarily imply strong dodging performance. When the target identity $y_t$ is absent from the attacker's model (e.g., simple verification against victim identity $y$), optimizing for ISR offers little privacy benefit. Moreover, impersonation methods introduce ethical and legal risks, as they enable deliberate framing of others (Wenger et al., 2023). By contrast, dodging misleads recognition in an uncontrolled manner—typically to an unknown or incorrect identity—thus preventing identification without enabling targeted misuse. For this reason, effective AFR for privacy should prioritize dodging.

More concretely, existing AFR techniques that excel in impersonation settings often suffer a sharp performance drop in DSR (Figure 1), exposing a fundamental gap: current methods are not optimized for dodging, leaving privacy protection unreliable when the user seeks only to avoid recognition. Even works that claim to support dodging (Shamshad et al., 2023; Fan et al., 2025) often depend on a target identity, contradicting the very goal of identity-free protection. Developing robust dodging-based AFR remains a key challenge for enabling reliable and ethically sound privacy defenses.

**Limits in Spatial Precision & User Control.** Noise-based approaches like TIP-IM produce global pixel-wise perturbations that are often visually perceptible, while GAN-based methods, such as AMT-GAN and C2P, introduce artifacts extending beyond the face area. This lack of spatial precision is problematic when users wish to disguise only their facial identity while preserving natural backgrounds, as shown in Figure 1. DiffAM improves localization by applying makeup perturbations from reference images, but its reliance on such references restricts flexibility when suitable examples are unavailable. Beyond localization, current methods also suffer from limited prompt-following ability, constraining user control and customization. Reference-based AFR struggles with consistent style transfer: early GAN methods (e.g., AMT-GAN) fail to replicate makeup reliably, while DiffAM remains restricted to three fixed facial regions (skin, mouth, eyes), excluding areas like eyebrows, and requires fine-tuning for each reference image. Text-prompt-based AFR removes this dependency and improves usability, but models like C2P, which rely solely on CLIP directional loss, often apply unintended modifications outside the target regions. Together, these issues highlight the need for AFR techniques that are both precisely localized and responsive to user intent.

### 4.2 DETAILED DESIGN OF MASQUE

To address the limitations of existing AFR methods discussed in Section 4.1, we propose **MASQUE**, a method designed to disrupt FR models with localized adversarial makeup while ensuring no external identity is introduced. Figure 2 illustrates the pipeline of **MASQUE** with its pseudocode detailed in Algorithm 1 in Appendix A. The makeup generation process is guided by a user-defined text prompt $p^*$, refined via cross-attention fusion with a mask. Pairwise adversarial guidance is further introduced to ensure these perturbations mislead FR models without compromising visual fidelity. In particular, our method leverages the state-of-the-art Stable Diffusion (Rombach et al., 2022) adapted to generate

high-quality adversarial makeup by iterative denoising a random noise latent conditioned on a text embedding $c$. To be more specific, Stable Diffusion is trained to predict the added noise $\epsilon$ via:

$$\min_\theta \; \mathbb{E}_{z_0,\epsilon\sim\mathcal{N}(\mathbf{0},\mathrm{I}),t\sim\mathrm{Unif}(1,T)} \left[ \left\| \epsilon - \epsilon_\theta(z_t, t, c) \right\|_2^2 \right]. \tag{3}$$

**Applying Makeup with Cross Attention.** Before applying edits, we first obtain a faithful latent representation of the original image $x$ using null-text inversion (Mokady et al., 2023), which mitigates reconstruction errors common in direct DDIM inversion (Song et al., 2020). Conditioning on an empty prompt to align the forward and reverse diffusion trajectories, ensuring near-perfect reconstruction of the image's structure and identity. With this accurate latent representation, we introduce makeup attributes via the text prompt $p^*$ by manipulating the cross-attention (CA) layers of the diffusion model (Rombach et al., 2022), which control how spatial features correspond to semantic tokens. At each diffusion step $\tau$, we extract attention maps $A_\tau$ (reconstruction) and $A_\tau^*$ (editing) and blend them: preserving CA values from $A_\tau$ for shared tokens to maintain structure, while incorporating values from $A_\tau^*$ for makeup-specific tokens in $p^*$:

$$\left(\mathrm{Update}(A_\tau, A_\tau^*)\right)_{i,j} := \begin{cases} (A_\tau)_{i,j}, & \text{if } j \text{ is in both } p \text{ and } p^*, \\ (A_\tau^*)_{i,j}, & \text{if } j \text{ is unique to } p^*, \end{cases} \tag{4}$$

where $p$ denotes the original text prompt. The result is $\hat{A}_\tau$, a set of mixed CA maps that preserve the original facial layout while steadily introducing adversarial makeup features (Hertz et al., 2022).

**Enhancing Semantic Edits & Locality.** To ensure precise localization, we generate a mask $\mathcal{M}$ that defines the region for modification. To achieve this, we embed the prompt $p^*$ using a Sentence Transformer (Reimers & Gurevych, 2019) model and compare it to embeddings of predefined facial regions. The closest match determines the relevant area for the edit. For instance, if the prompt specifies "a face with red lipstick", the model identifies lips as the target and generates a lip-area mask. This ensures that our adversarial makeup perturbations remain focused on the correct facial region and do not unintentionally alter other parts of the face. Once the target region $\mathcal{M}$ is determined, we enhance the influence of makeup-related tokens by maximizing their attention within $\mathcal{M}$. This ensures that modifications are reinforced within the intended area, preventing unintended changes elsewhere and maintaining overall image quality (Mao et al., 2023). Specifically, we optimize:

$$\mathcal{L}_{\text{edit}} = \left( 1 - \frac{1}{|\mathcal{M}|} \sum_{i \in \mathcal{M}} \frac{(A_\tau^*)_{i,\text{new}}}{(A_\tau^*)_{i,\text{new}} + \sum_{j \in \text{share}}(A_\tau)_{i,j}} \right)^2, \tag{5}$$

where $(A_\tau^*)_{i,\text{new}}$ is the attention weights assigned to the new makeup tokens at spatial index $i$, while $\sum_{j \in \text{share}}(A_\tau)_{i,j}$ is the total attention weight of tokens that appear in both the original and makeup prompts. By emphasizing makeup-specific attention in the masked region, this loss term ensures that modifications in $x_{\text{p}}$ occur meaningfully in the intended region while preventing edits from spreading elsewhere. To enforce spatial precision, we process latents through two distinct branches: an edit branch for makeup application and a reconstruction branch to preserve original features. During the backward steps, we explicitly constrain perturbations within the designated area by imposing:

$$z_{\text{edit}} = \mathcal{M} \cdot z_{\text{edit}} + (1 - \mathcal{M}) \cdot z_{\text{rec}}, \tag{6}$$

where $z_{\text{edit}}$ denotes the latent from the edit branch, and $z_{\text{rec}}$ is the latent from the reconstruction branch. While makeup edits ensure semantic plausibility, the core adversarial objective is to disrupt FR models. Therefore, we propose a novel *pairwise adversarial guidance* loss that uses a guide image of the same identity to achieve robust identity confusion without relying on an external target.

**Pairwise Adversarial Guidance.** Previous makeup-based AFR methods often target another identity, compromising privacy and limiting applicability in dodging scenarios. By contrast, our approach leverages a pair $(x, \tilde{x})$ of face images from the same individual, where $\tilde{x}$ serves as the guide image. This strategy highlights a significant issue with using only the distance from the original image as the adversarial loss. Diffusion models are designed to reconstruct images similar to the original, so maximizing distance alone creates conflicting objectives. As a result, this approach often produces unstable performance in both image quality and adversarial effectiveness. In our evaluations, we also study the impact of the number of guide images $G$ on MASQUE's performance (Appendix C.1)

Table 2: DSR (%) of various AFR methods under facial identification and verification settings. Here, $G$ denotes the number of guide images used in MASQUE. Each best result is highlighted in bold.

| | Method | CelebA-HQ | | | | VGG-Face2-HQ | | | | Avg. |
|---|---|---|---|---|---|---|---|---|---|---|
| | | IR152 | IRSE50 | FaceNet | MobileFace | IR152 | IRSE50 | FaceNet | MobileFace | |
| Identification | Clean | 10.0 | 13.0 | 5.0 | 40.0 | 13.0 | 18.0 | 18.0 | 25.0 | 17.8 |
| | TIP-IM | 62.0 | 86.0 | 65.0 | 74.0 | 57.0 | 73.0 | 52.0 | 62.0 | 66.4 |
| | AMT-GAN | 61.2 | 48.3 | 50.7 | 57.3 | 16.0 | 24.0 | 22.3 | 32.3 | 39.0 |
| | DiffAM | 54.0 | 57.7 | 59.0 | 74.3 | 49.7 | 51.7 | 54.0 | 70.7 | 58.9 |
| | C2P (I) | 30.7 | 38.3 | 22.0 | 56.7 | 18.00 | 20.0 | 21.7 | 32.0 | 29.9 |
| | C2P (D) | 74.7 | 76.3 | 56.3 | 77.3 | 18.3 | 19.7 | 20.3 | 30.3 | 46.7 |
| | FPP | 69.0 | 74.0 | 75.0 | 55.0 | 46.0 | 54.0 | 73.0 | 34.0 | 60.0 |
| | MASQUE ($G = 0$) | 92.3 | 95.7 | 61.3 | 87.0 | 69.7 | 76.3 | 45.3 | 81.7 | 76.2 |
| | MASQUE ($G = 1$) | **98.0** | **98.3** | **77.7** | **94.0** | **84.3** | **89.0** | **61.0** | **91.0** | **85.5** |
| Verification | Clean | 5.0 | 5.0 | 4.0 | 10.0 | 13.0 | 17.0 | 15.0 | 30.0 | 12.4 |
| | TIP-IM | 44.0 | 60.0 | 51.0 | 40.0 | 46.0 | 51.0 | 59.0 | 46.0 | 49.6 |
| | AMT-GAN | 40.0 | 32.0 | 51.7 | 25.7 | 14.0 | 26.3 | 26.7 | 40.7 | 32.1 |
| | DiffAM | 31.0 | 26.0 | 54.0 | 43.3 | 46.3 | 50.7 | 56.0 | 82.7 | 48.8 |
| | C2P (I) | 12.3 | 11.3 | 14.3 | 21.3 | 16.3 | 18.7 | 19.3 | 37.3 | 18.9 |
| | C2P (D) | 52.3 | 46.6 | 40.3 | 41.7 | 17.0 | 19.7 | 18.3 | 38.0 | 34.3 |
| | FPP | 42.0 | 40.0 | 41.0 | 53.0 | 44.0 | 48.0 | 81.0 | 34.0 | 47.8 |
| | MASQUE ($G = 0$) | 89.3 | 90.7 | 57.7 | 76.3 | 62.3 | 71.3 | 42.3 | 86.0 | 72.0 |
| | MASQUE ($G = 1$) | **96.0** | **96.7** | **75.7** | **79.3** | **82.3** | **91.7** | **63.3** | **94.3** | **84.9** |

and whether using self-augmentation without a separate guide image ($G = 0$) can be a competitive alternative (Appendix C.2).

**Balancing Protection Strength & Image Quality.** We introduce adversarial perturbations during the later stage of the diffusion process, ensuring the coarse structure remains intact while subtly altering identity-specific features. To balance adversarial potency with visual fidelity, we incorporate perceptual similarity constraints alongside a *cosine similarity* (CoSi) measure:

$$\mathcal{L}_{adv} = \lambda_{CoSi} \cdot CoSi(\boldsymbol{x}_p, \tilde{\boldsymbol{x}}) + \lambda_{LPIPS} \cdot LPIPS(\boldsymbol{x}_p, \boldsymbol{x}), \tag{7}$$

where $\boldsymbol{x}_p, \tilde{\boldsymbol{x}}$ refer to the protected and guide images, and $\lambda_{CoSi}, \lambda_{LPIPS}$ are trade-off parameters. Here, $CoSi(\boldsymbol{x}_p, \tilde{\boldsymbol{x}})$ ensures that the perturbations sufficiently diverge from recognizable identity features, while the LPIPS loss (Zhang et al., 2018) maintains perceptual and structural fidelity.

## 5 EXPERIMENTS

### 5.1 EXPERIMENTAL SETUP

To reflect the high-quality nature of online face images, we use 300 high-resolution images ($1024 \times 1024$) from each of the CelebA-HQ (Karras et al., 2017) and VGG-Face2-HQ (Chen et al., 2024) datasets. For each dataset, we randomly sample 100 identities, each with three images: a probe $\boldsymbol{x}$ (to be protected), a reference $\tilde{\boldsymbol{x}}$ (as the guide image), and a gallery image $\boldsymbol{x}_g$ (stored in the attacker's facial recognition model). Note that the VGG-Face2-HQ dataset contains face images under challenging conditions, such as pose variation and natural occlusions like sunglasses and hair.

We compare MASQUE against several baselines: TIP-IM (Yang et al., 2021), AMT-GAN (Hu et al., 2022), C2P (Shamshad et al., 2023), DiffAM (Sun et al., 2024), and FPP (Salar et al., 2025). For C2P, both its impersonating version, C2P (I), and its dodging version, C2P (D), are assessed. We compare the performance of MASQUE with existing AFR techniques on four public FR models: IR152 (Deng et al., 2019), IRSE50 (Hu et al., 2018), FaceNet (Schroff et al., 2015), and MobileFace (Chen et al., 2018). Without explicitly mentioning, we set $G = 1$ in MASQUE (using a single separate guide image of the same individual). For reference-based methods, we selected images from the makeup dataset (Li et al., 2018) used during their pre-training that best matched the given prompt. In addition to public FR models, we evaluate AFR methods against two commercial APIs: Face++[1] and Luxand[2].

For evaluation, we use DSR to measure the protection strength of AFR methods and LPIPS, PSNR, and SSIM (Wang et al., 2004) to assess visual quality, where lower LPIPS and higher PSNR or SSIM

---

[1] https://www.faceplusplus.com/face-comparing/
[2] https://luxand.cloud/face-api

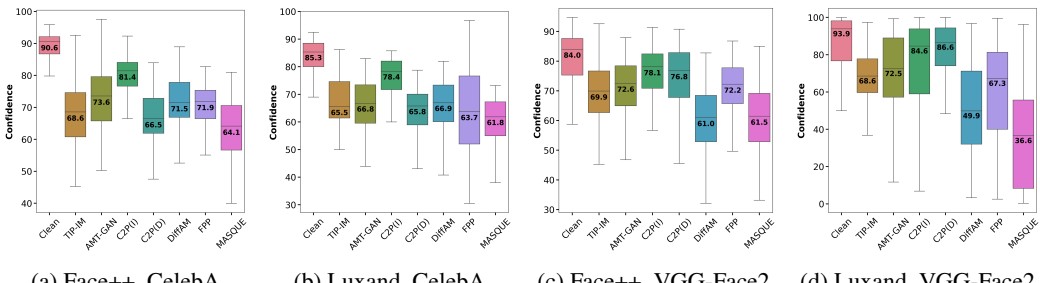

| (a) Face++, CelebA | (b) Luxand, CelebA | (c) Face++, VGG-Face2 | (d) Luxand, VGG-Face2 |

Figure 3: Performance comparisons between different AFR methods against two commercial APIs.

Table 3: Visual quality of various AFR methods on CelebA-HQ averaged across three makeup styles.

|  | TIP-IM | AMT-GAN | C2P(I) | C2P(D) | DiffAM | FPP | MASQUE ($G = 0$) | MASQUE ($G = 1$) |
|---|---|---|---|---|---|---|---|---|
| LPIPS ($\downarrow$) | 0.311 | 0.342 | 0.460 | 0.473 | 0.399 | 0.456 | 0.298 | 0.294 |
| PSNR ($\uparrow$) | 32.16 | 19.51 | 18.92 | 17.99 | 18.31 | 20.53 | 25.33 | 25.82 |
| SSIM ($\uparrow$) | 0.928 | 0.613 | 0.583 | 0.563 | 0.769 | 0.595 | 0.839 | 0.856 |

indicate better preservation. Without explicit mention, the results are averaged across three makeup prompts, "red lipstick", "blue eyebrow" and "pink eyeshadow". We also evaluate prompt adherence with metrics such as concentration and hue accuracy. Appendix B provides more details regarding the hyperparameters of `MASQUE`, external reference implementation, and metric definitions. Additional experiments, ablations, and visualizations are provided in Appendices C–E.

## 5.2 MAIN RESULTS

**High Dodging Success.** Table 2 presents the DSR for face verification and rank-1 identification in black-box settings using four widely adopted pre-trained FR extractors. For each target model, the others serve as surrogates, with results averaged across three makeup styles. Table 2 demonstrates that `MASQUE` with $G = 1$ consistently outperforms existing baselines across all configurations, achieving an average DSR of $85.5\%$ for identification and $84.9\%$ for verification. Besides, even without a guide image, `MASQUE` still outperforms TIP-IM, the second-best AFR method, by a large margin, highlighting strong protection under the dodging scenario. We also test our method against two commercial FR APIs in verification mode, which assigns similarity scores from 0 to 100. As proprietary models with unknown training data and parameters, they provide realistic testbeds for our method. Figure 3 shows that our method achieves the lowest similarity scores across both APIs, confirming effectiveness in open- and closed-source settings and reinforcing real-world applicability.

**Visual Quality Preservation.** `MASQUE` achieves superior image quality across multiple evaluation metrics, as summarized in Table 3. While TIP-IM attains the highest PSNR and SSIM due to its small perturbation constraint, these pixel-level metrics often fail to reflect perceptual quality (see Figure 1 for qualitative comparison results). In contrast, our approach prioritizes perceptual consistency, balancing content fidelity and visual realism, as demonstrated by its strong LPIPS performance.

**Localization & Prompt Adherence.** Moreover, we examine the localization capability and prompt adherence of `MASQUE` based on spatial precision and color accuracy. For spatial adherence, we use *concentration*, which quantifies the proportion of modifications within a binary mask of the target area. Higher scores indicate better localization, while lower scores suggest spillover. For color precision, we evaluate *hue accuracy* by extracting modified pixels within the mask, converting them to HSV space, and measuring how many fall within the expected hue range. Figure 4 shows that `MASQUE` consistently achieves the highest concentration scores and competitive hue accuracy across prompts, ensuring precise localization with accurate color application. In contrast, DiffAM struggles with blue eyebrows, likely due to its framework restricting makeup transfer to specific facial regions, limiting its flexibility. In Appendix D.1, we conduct additional experiments using other distance metrics with respect to both in- and out-mask regions, illustrating the strong localization capability of `MASQUE`. These results confirm that `MASQUE` achieves strong prompt adherence by maintaining spatial precision while accurately reflecting the intended color attributes.

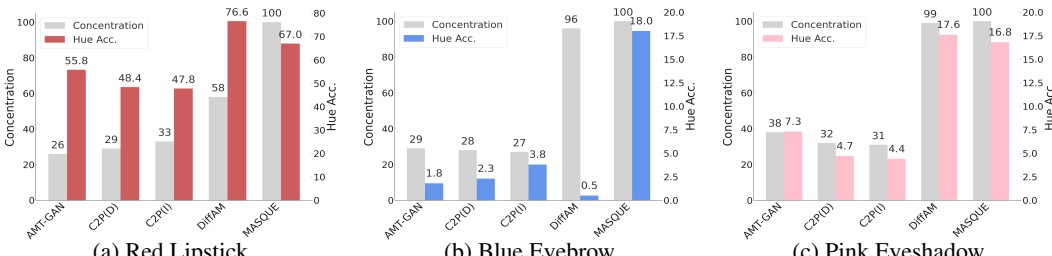

(a) Red Lipstick      (b) Blue Eyebrow      (c) Pink Eyeshadow

Figure 4: Comparison of concentration and hue accuracy on CelebA-HQ across different AFR methods for three makeup styles. Concentration values are normalized relative to `MASQUE`.

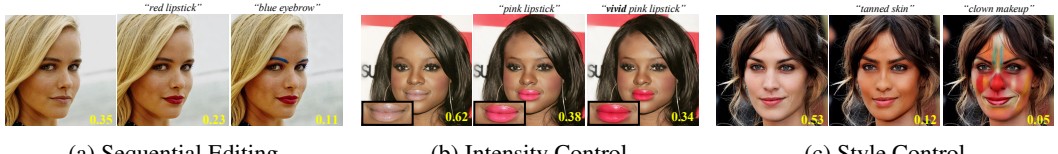

(a) Sequential Editing      (b) Intensity Control      (c) Style Control

Figure 5: Prompt adherence of `MASQUE`, where similarity score is shown at the bottom right.

Table 4: DSR of `MASQUE` with "red lipstick" on CelebA-HQ under various image transformations.

|  | None | Resize | Compression | Gaussian | Blur | Fog | De-Makeup | Denoising |
|---|---|---|---|---|---|---|---|---|
| Identification | 0.905 | 0.885 | 0.898 | 0.925 | 0.898 | 0.933 | 0.913 | 0.800 |
| Verification | 0.818 | 0.790 | 0.793 | 0.898 | 0.813 | 0.885 | 0.760 | 0.778 |

## 6 FURTHER ANALYSES

**Controllable Makeup Generation.** Figure 5 illustrates the strong controllability of `MASQUE` over text makeup prompts across three specific dimensions: sequential editing, intensity control, and style control. Makeup can be applied progressively to multiple facial regions, with each layer enhancing protection, as shown by decreasing similarity scores. Intensity is adjustable via text prompts (e.g., "vivid"), affecting appearance with minimal impact on effectiveness. Style prompts range from natural (e.g., "tanned skin") to extreme (e.g., "clown makeup"), with broader coverage offering stronger protection (see Appendix E for additional visualizations of `MASQUE`-generated images).

**Robustness to Image Transformation.** We evaluate the robustness of `MASQUE` using the same 100 CelebA-HQ images and four FR models used in the main experiments, with the prompt "red lipstick". We then subject the edited images to a variety of transformations, standard post-processing, such as resize and JPEG compression, natural image corruptions, such as Gaussian noise, blur, and fog, and adaptive variations such as DiffAM's makeup removal and DDPM-based denoising. Table 4 summarizes the averaged DSR across the four considered public FR models, where Table 11 in the appendix provides the full comparison results. `MASQUE` maintains high dodging success rates across all cases, indicating strong resilience to real-world degradations and adaptive threats. This robustness may arise from the invariance of FR models, which are trained to withstand such transformations, allowing `MASQUE`'s protection to persist. Also, we test the performance of `MASQUE` against adversarial purification (Nie et al., 2022), where the results are presented and discussed in Appendix D.4.

## 7 CONCLUSION

We introduced `MASQUE`, a diffusion-based AFR framework that enables localized adversarial makeup synthesis via text prompts. Unlike prior AFR methods, `MASQUE` achieves high DSRs while preserving visual quality without relying on external images, reducing the potential risks of targeted misuse. Our experiments demonstrated that `MASQUE` outperforms existing baselines, with pairwise adversarial guidance playing a central role. Future work may focus on optimizing the efficiency for real-time applications and studying the robustness against adaptively evolving FR models (Fan et al., 2025) to further strengthen `MASQUE`'s efficacy for facial privacy protection.

## ETHICS STATEMENT

Our work only uses public datasets and aims to defend against unauthorized or invasive FR systems, so we believe it does not raise any direct ethical concerns. Unlike prior AFR works that are mostly impersonation-driven, our method is distinctive in that it avoids using any external identity's face image, thereby reducing the risks of targeted abuse of the technology. Nevertheless, our work may raise dual-use concerns. In particular, the same techniques can potentially be misused to evade legitimate or authorized FR systems, such as those used in security or law enforcement contexts. It is important to anticipate and acknowledge these risks, and we encourage future work to consider safeguards or contextual constraints that limit misuse while preserving individual privacy protections.

## REPRODUCIBILITY STATEMENT

To ensure reproducibility, we provide the details of our implementations, including pseudocode in Appendix A and hyperparameter setup in Appendix B. For datasets, we use public image benchmarks, CelebA-HQ and VGG-Face2, which naturally suggest reproducibility. Besides, our implementations are available as anonymized code at https://anonymous.4open.science/r/MASQUE-54C2.

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

# A PSEUDOCODE OF MASQUE

Algorithm 1 presents the pseudocode of `MASQUE`, which consists of modules such as precise null-text inversion, cross-attention fusion with masking, and pairwise adversarial guidance.

---

**Algorithm 1** Adversarial Makeup Generation with Pairwise Adversarial CA Guidance

---

**Input:** $z_{\text{edit}}, z_{\text{rec}}$: edited and original latent, $e_{\text{edit}}, e_{\text{rec}}$: edited and original text embeddings, $\mathcal{D}$: Stable Diffusion model, $T$: number of diffusion steps, $\tau_{\text{attn}}, \tau_{\text{edit}}, \tau_{\text{adv}}$: step thresholds, $\mathcal{M}$: binary region mask, $\mathcal{I}_{\text{target}}$: target token indices, $d_k$: key/query dimensionality, $\lambda_{\text{edit}}, \lambda_{\text{CoSi}}, \lambda_{\text{LPIPS}}$: loss weights, $\eta$: learning rate, $\sigma(t)$: noise scale function, $x$: input image, $\tilde{x}$: guide image, $m_{\text{adv}}$: maximum adversarial iterations

**Output:** Final protected image $x_{\text{p}}$

**for** $k \leftarrow T$ **to** $1$ **do**

  **if** $k > T - \tau_{\text{attn}}$ **then**
    //Perform cross-attention edit
    $\text{CA}^{\text{refined}} \leftarrow \{\}$
    **for** $l \in \textit{CrossAttnLayers}$ **do**
      $Q_l, K_l, V_l \leftarrow \mathcal{D}.\text{UNet}.\text{GetCrossAttentionComponents}(z_{\text{edit}}, e_{\text{edit}}, l)$
      $\text{CA}_l \leftarrow \text{Softmax}\left(\frac{Q_l\,K_l^\top}{\sqrt{d_k}}\right)$
      $\text{CA}_l^{\text{refined}} \leftarrow \text{CA}_l \cdot V_l$
      $\text{CA}^{\text{refined}} \leftarrow \text{CA}^{\text{refined}} \cup \{\text{CA}_l^{\text{refined}}\}$

  **if** $k > T - \tau_{\text{edit}}$ **then**
    //Perform edit loss update
    $\mathcal{L}_{\text{edit}} \leftarrow 0$
    **foreach** $\text{CA}_l^{\text{refined}} \in \text{CA}^{\text{refined}}$ **do**
      $\text{CA}_{\text{target}} \leftarrow \text{ExtractTargetAttention}\left(\text{CA}_l^{\text{refined}}, \mathcal{I}_{\text{target}}\right)$
      $\text{CA}_{\text{masked}} \leftarrow \text{CA}_{\text{target}} \odot \mathcal{M}$
      $\mathcal{L}_{\text{edit}} \leftarrow \mathcal{L}_{\text{edit}} + \frac{\left(\sum_{(i,j)\in\mathcal{M}} \text{CA}_{\text{masked}}[i,j]\right)^2}{\sum_{(i,j)\in\mathcal{M}} \mathcal{M}[i,j]}$
    $\mathcal{L}_{\text{edit}} \leftarrow \frac{\mathcal{L}_{\text{edit}}}{|\text{CA}^{\text{refined}}|}$
    $\nabla_{z_{\text{edit}}} \mathcal{L}_{\text{edit}} \leftarrow \text{Backprop}(\mathcal{L}_{\text{edit}})$
    $z_{\text{edit}} \leftarrow z_{\text{edit}} - \eta \cdot \lambda_{\text{edit}} \cdot \nabla_{z_{\text{edit}}} \mathcal{L}_{\text{edit}}$

  //Apply classifier-free guidance during all diffusion steps
  $\epsilon_{\text{rec}} \leftarrow \mathcal{D}.\text{UNet}(z_{\text{rec}}, t_k, e_{\text{rec}})$
  $\epsilon_{\text{edit}} \leftarrow \mathcal{D}.\text{UNet}(z_{\text{edit}}, t_k, e_{\text{edit}}, \text{CA}^{\text{refined}})$
  $z_{\text{rec}} \leftarrow z_{\text{rec}} - \sigma(t_k) \cdot \epsilon_{\text{rec}}$
  $z_{\text{edit}} \leftarrow z_{\text{edit}} - \sigma(t_k) \cdot \epsilon_{\text{edit}}$
  $z_{\text{edit}} = \mathcal{M} \cdot z_{\text{edit}} + (1 - \mathcal{M}) \cdot z_{\text{rec}}$

  **if** $k < T - \tau_{\text{adv}}$ **then**
    //Perform pairwise adversarial optimization
    **for** $i \leftarrow 0$ **to** $m_{\text{adv}} - 1$ **do**
      $x_{\text{rec}} \leftarrow \mathcal{D}.\text{VAE}.\text{Decode}(z_{\text{edit}})$
      $\mathcal{L}_{\text{adv}} \leftarrow \lambda_{\text{CoSi}} \cdot \text{CoSi}\big(\text{FR}(x_{\text{rec}}), \text{FR}(\tilde{x})\big) + \lambda_{\text{LPIPS}} \cdot \text{LPIPS}(x_{\text{rec}}, x)$
      $\text{grad} \leftarrow \text{Backprop}(\mathcal{L}_{\text{adv}})$
      $z_{\text{edit}} \leftarrow z_{\text{edit}} - \eta \cdot \text{grad}$
      $z_{\text{edit}} \leftarrow z_{\text{edit}} \odot \mathcal{M} + z_{\text{rec}} \odot (1 - \mathcal{M})$

//Decode the final latent to produce the protected image
$x_{\text{p}} \leftarrow \mathcal{D}.\text{VAE}.\text{Decode}(z_{\text{edit}})$
**return** $x_{\text{p}}$

---

Table 5: Comparison of identification confidence, verification similarity, and image quality metrics across different numbers of guide images employed in `MASQUE`.

| # Guide Images | Iden. DSR ($\uparrow$) | Veri. DSR ($\uparrow$) | Confidence ($\uparrow$) | Similarity ($\downarrow$) | LPIPS ($\downarrow$) |
|---|---|---|---|---|---|
| 0 | 0.66 | 0.60 | 0.114 | 0.355 | 0.303 |
| 1 | 0.67 | 0.63 | 0.135 | 0.349 | 0.292 |
| 2 | 0.72 | 0.71 | 0.165 | 0.297 | 0.291 |
| 5 | 0.74 | 0.74 | 0.177 | 0.283 | 0.291 |
| 10 | 0.76 | 0.77 | 0.197 | 0.266 | 0.291 |

## B  DETAILED EXPERIMENTAL SETTINGS

**Implementation Details.** `MASQUE` builds on the pre-trained Stable Diffusion v1.4 model (Rombach et al., 2022), using a denoising diffusion implicit model (DDIM) (Song et al., 2020) denoising over $T = 50$ steps with a fixed guidance scale of 7.5. During backward diffusion, CA injection occurs in $[T, T - \tau_{\text{attn}}]$ ($\tau_{\text{attn}} = 40$), localize optimization in $[T, T - \tau_{\text{edit}}]$ ($\tau_{\text{edit}} = 5$), and adversarial guidance in $[T - \tau_{\text{adv}}, 0]$ ($\tau_{\text{adv}} = 45$). We set $\lambda_{\text{CoSi}} = 0.1$, $\lambda_{\text{LPIPS}} = 1$, and cap optimization to $m_{\text{adv}} = 15$ iterations. We select the value of these hyperparameters based on the prior literature or our ablation studies presented in Appendix C.3.

**Reference Images.** For reference-based methods, we selected images from the makeup dataset (Li et al., 2018) used during their pre-training that best matched the given prompt. For the "blue eyebrow" style—introduced to assess performance on an uncommon makeup—finding a suitable reference was challenging, underscoring the limitations of reference-based methods for rare styles. To address this, we inpainted blue eyebrows on a non-makeup image from the dataset using our framework, excluding pairwise adversarial guidance, showing its capacity for targeted edits without an external reference.

**Evaluation Metrics.** We use the dodging success rate (DSR) as the primary metric for evaluating the protection effectiveness of AFR. DSR is computed using a thresholding strategy for face verification and a closed-set strategy for face identification. For verification, DSR is defined as:

$$\text{DSR}_{\text{FV}} = \frac{1}{|\mathcal{D}|} \sum_{(\boldsymbol{x}, y) \in \mathcal{D}} \mathbb{1}\big( \cos(f(\boldsymbol{x}_{\text{p}}), f(\boldsymbol{x}_{\text{g}})) > \gamma \big),$$

where $f$ denotes the feature extractor of the FR model, and $\boldsymbol{x}_{\text{g}}$ stands for the gallery image of the victim identity. The similarity threshold $\gamma$ is set at a 0.01 *false acceptance rate* (FAR) for each FR model. For face identification, the DSR of an AFR method is defined as the optimization objective of Equation 1. In particular, we use the rank-1 accuracy, which measures whether the top-1 candidate list excludes the original identity corresponding to the victim user's image $\boldsymbol{x}$.

## C  ABLATION STUDIES

In this section, we present detailed ablation studies to comprehensively examine the impact of different components of our approach. All experiments are conducted on the CelebA-HQ dataset, using the FaceNet model as the target and "red lipstick" as the makeup prompt.

### C.1  NUMBER OF IMAGES FOR ADVERSARIAL GUIDANCE

`MASQUE` introduces pairwise adversarial guidance to protect identities without external target references, distinguishing it from impersonation-oriented methods. By aligning features between the guide and original images, we identify identity-relevant cues and inject subtle adversarial signals that steer the diffusion process away from a recognizable identity manifold. The guide image serves as a proxy for the gallery image, providing guidance on the direction in which the latent representation should be altered. This eliminates reliance on external identities, mitigating ethical concerns related to impersonation while ensuring identity protection remains non-intrusive.

Using multiple guide images further stabilizes training, reducing bias toward any single representation and improving robustness. To evaluate its effectiveness, we use FaceNet as the target model and select 100 identities from CelebA-HQ, varying the number of guide images per identity. These

Table 6: Impact of data augmentation on the guide image of `MASQUE` on DSR and PSNR.

| Method | # Guide Images | Augmentation | Iden. DSR ($\uparrow$) | Veri. DSR ($\uparrow$) | PSNR ($\uparrow$) |
|---|---|---|---|---|---|
| Baseline | $G = 1$ | none | 0.72 | 0.66 | 25.69 |
| Baseline | $G = 0$ | none | 0.56 | 0.40 | 25.81 |
| Self-aug. | $G = 0$ | horizontal flip | 0.57 | 0.50 | 25.83 |
| Self-aug. | $G = 0$ | vertical flip | 0.12 | 0.06 | 26.57 |
| Self-aug. | $G = 0$ | translation | 0.55 | 0.38 | 25.81 |

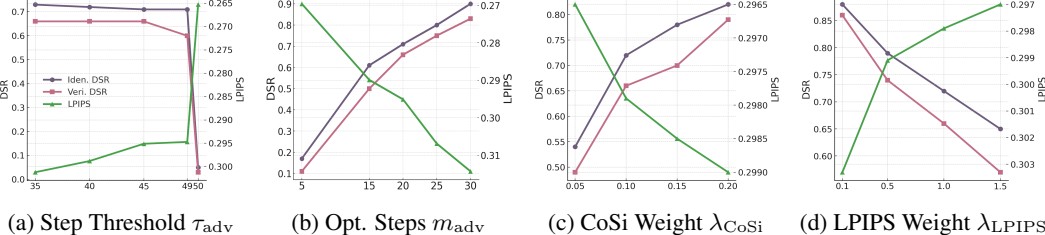

(a) Step Threshold $\tau_{\mathrm{adv}}$    (b) Opt. Steps $m_{\mathrm{adv}}$    (c) CoSi Weight $\lambda_{\mathrm{CoSi}}$    (d) LPIPS Weight $\lambda_{\mathrm{LPIPS}}$

Figure 6: Ablations on `MASQUE` showing trade-off between protection strength and image quality.

evaluation identities are distinct from the main experiments, ensuring at least 12 images per identity to accommodate setups with up to 10 guide images. Effectiveness is measured using the confidence score, the gap between the similarity score of the original identity's gallery image and that of the misidentified identity. Table 5 shows that increasing guide images enhances identity obfuscation, leading to higher confidence scores while reducing similarity with the original identity. Moreover, images generated with guide images in the pairwise adversarial guidance step exhibit higher visual quality than those relying solely on adversarial loss on the original image. This confirms that simply maximizing distance from the original image is ineffective, as it disrupts the diffusion process's goal of preserving natural image characteristics, leading to instability in both image quality and adversarial performance. These results demonstrate that `MASQUE`'s pairwise adversarial guidance provides a robust, privacy-centric solution, balancing strong identity obfuscation with high visual fidelity.

### C.2 SELF-AUGMENTED GUIDANCE

To address cases where users may lack suitable guide images, we tested using self-augmented input images. For augmentation, we applied horizontal flipping, vertical flipping, and random translation with a maximum shift of 20 pixels. Results, as shown in Table 6, compare these approaches against the input image itself as the guide image. Horizontal flipping and random translation perform similarly to using the input image as the guide, whereas vertical flipping shows minimal adversarial effect. This is likely because vertically flipped faces are hard to align with facial images for the recognition model.

### C.3 ADVERSARIAL STRENGTH VERSUS IMAGE QUALITY

Figure 6 presents our ablation studies on the impact of hyperparameters used in Algorithm 1: $\tau_{\mathrm{adv}}$, $m_{\mathrm{adv}}$, $\lambda_{\mathrm{CoSI}}$, and $\lambda_{\mathrm{LPIPS}}$. Increasing the value of $\tau_{\mathrm{adv}}$ enhances the adversarial effect, improving DSR but degrading image quality, as indicated by higher LPIPS values. Similarly, increasing the number of pairwise adversarial optimization steps $m_{\mathrm{adv}}$ amplifies protection but further compromises image quality. According to Figures 6c and 6d, we select $\lambda_{\mathrm{CoSi}} = 0.1$, $\lambda_{\mathrm{LPIPS}} = 1.0$ to balance the trade-off between the dodging success rates and the visual quality of generated images.

### C.4 MASK AND PERTURBATION LOCALIZATION

To understand the impact of masking, we further conduct an ablation study with and without applying a mask in the proposed framework of `MASQUE`. Figure 7 shows that the absence of a mask causes the model to struggle with spatial control, leading to unconstrained perturbations across the entire image rather than confining makeup to the intended area.

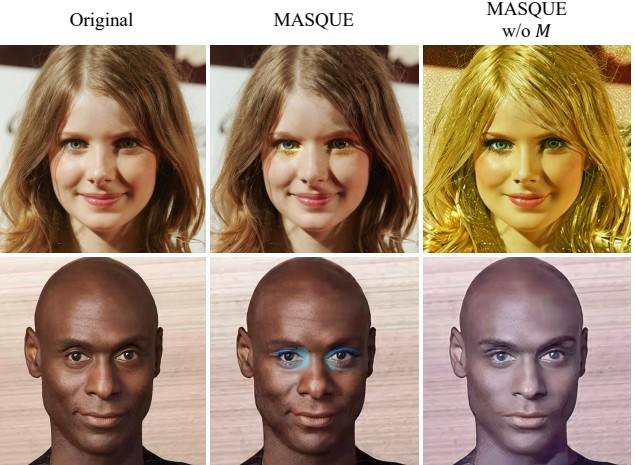

Figure 7: Visualizations of MASQUE ($G = 1$) with and without masking. The first row uses prompt "shimmery gold eyeshadow", while the second row uses "blue eyeshadow".

Table 7: Comparison of similarity metrics between AFR methods across in- and out-mask regions.

| Metric | Type | AFR Method | | | | | |
|---|---|---|---|---|---|---|---|
| | | TIP-IM | AMT-GAN | C2P (I) | C2P (D) | DiffAM | MASQUE ($G = 1$) |
| DISTS ($\downarrow$) | In-Mask | 2.013 | 7.993 | 8.945 | 10.011 | 9.645 | 12.306 |
| | Out-Mask | 0.094 | 0.149 | 0.153 | 0.169 | 0.141 | 0.104 |
| | Diff ($\Delta$) | 1.919 | 7.844 | 8.792 | 9.842 | 9.504 | 12.202 |
| LPIPS ($\downarrow$) | In-Mask | 0.322 | 0.360 | 0.400 | 0.4356 | 0.509 | 0.434 |
| | Out-Mask | 0.305 | 0.316 | 0.447 | 0.460 | 0.410 | 0.290 |
| | Diff ($\Delta$) | 0.017 | 0.044 | $-0.047$ | $-0.025$ | 0.099 | 0.144 |
| PieAPP ($\downarrow$) | In-Mask | 3.621 | 21.325 | 17.186 | 20.850 | 17.523 | 21.434 |
| | Out-Mask | 0.462 | 0.813 | 1.138 | 1.314 | 1.312 | 0.645 |
| | Diff ($\Delta$) | 3.159 | 20.511 | 16.049 | 19.536 | 16.211 | 20.789 |

# D  ADDITIONAL EXPERIMENTS AND DISCUSSIONS

## D.1  LOCALIZED EDITING

To evaluate whether edits are confined to the desired region, images are divided into in-mask and out-mask regions using a binary mask. The corresponding areas in both images are isolated via element-wise multiplication. The areas are then used to normalize similarity metrics proportionally, ensuring fair comparisons. Specifically, we employ DISTS, LPIPS, and PieAPP as evaluation metrics in this experiment: DISTS measures perceptual dissimilarity based on structure and texture, LPIPS uses deep features, and PieAPP reflects human perceptual preferences.

Table 7 compares the effectiveness of localized edits, where higher $\Delta$ values indicate stronger localization of perturbations. TIP-IM (Yang et al., 2021), as a noise-based method, applies pixel-wise adversarial perturbations uniformly, resulting in the smallest difference for all metrics due to minimal distinction between in-mask and out-mask regions. In contrast, our method introduces significant perturbations in the in-mask region, leading to poorer metrics there, but achieves the best or second-best results in the out-mask region, indicating minimal disruption to untouched areas.

## D.2  DSR UNDER RANK-5 ACCURACY

We also report DSR for the face identification task, using rank-5 accuracy to assess dodging success. The results are presented in Table 8. Notably, MASQUE achieves the highest DSR in both rank-1 and rank-5 settings, demonstrating its effectiveness in misleading identification models.

Table 8: DSR (%) in terms of rank-1 and rank-5 accuracy in identification mode across AFR methods.

| Method | AMT-GAN | DiffAM | C2P (I) | C2P (D) | MASQUE ($G = 1$) |
|---|---|---|---|---|---|
| Rank-1 Accuracy | 54.39 | 61.25 | 36.92 | 71.17 | 92.08 |
| Rank-5 Accuracy | 28.58 | 45.58 | 18.25 | 48.67 | 81.42 |

Table 9: Visual assessment of AFR methods on VGG-Face2-HQ averaged for three makeup styles.

| | TIP-IM | AMT-GAN | C2P (I) | C2P (D) | DiffAM | FPP | MASQUE ($G = 1$) |
|---|---|---|---|---|---|---|---|
| LPIPS ($\downarrow$) | 0.220 | 0.275 | 0.357 | 0.356 | 0.499 | 0.540 | 0.205 |
| PSNR ($\uparrow$) | 26.58 | 20.18 | 19.12 | 19.37 | 14.29 | 14.52 | 25.34 |
| SSIM ($\uparrow$) | 0.882 | 0.797 | 0.664 | 0.673 | 0.451 | 0.423 | 0.872 |

### D.3 VGG-FACE2 IMAGE QUALITY

We report image quality metrics on the VGG-Face2 dataset, averaged over four face recognition models and three makeup styles. The results are shown in Table 9. MASQUE demonstrates consistently strong performance, in line with the results observed on the CelebA-HQ dataset.

### D.4 ROBUSTNESS AGAINST ADVERSARIAL PURIFICATION

Table 10 assesses MASQUE against adversarial purification techniques (Nie et al., 2022). When the purifier is fine-tuned on CelebA-HQ, it successfully removes MASQUE-generated perturbations on images from the same dataset. However, its effectiveness drops significantly on images from VGGFace2-HQ, revealing limited generalization to unseen identities and domains. Moreover, the computational overhead of adversarial purification—especially for high-resolution inputs—makes it impractical for real-time face recognition systems, particularly in tracking or surveillance scenarios.

### D.5 COMPUTATIONAL ANALYSIS

MASQUE does not require fine-tuning, operating directly with a pretrained Stable Diffusion model. In contrast, Clip2Protect fine-tunes per image (~25s), and DiffAM performs per-dataset fine-tuning (~37min) for each style or identity. At inference, MASQUE runs a one-time iterative optimization (~90s), slightly slower than Clip2Protect (~18s) and DiffAM (~30s). Despite this, MASQUE offers key advantages: it supports arbitrary prompts and identities without retraining, making it more flexible and general. Under typical AFR scenarios, where protection is applied once before sharing, this adaptability outweighs the modest runtime difference.

### D.6 LIMITATIONS

MASQUE may fail when the mask is very small, as downscaling may shrink or remove it entirely. We applied iterative dilation during early diffusion steps to address this, but very small masks may still vanish. Moreover, real-world AFR threats are constantly evolving, with adversaries continuously refining their strategies (Fan et al., 2025). Incorporating adaptive threat models and adversarial fine-tuning into the evaluation process could improve robustness and lead to a more comprehensive assessment of defense performance.

## E ADDITIONAL VISUALIZATIONS

Additional visual results of image-guided makeup transfer are provided in Figure 8 and Figure 9 to accommodate space limitations in the main text. MASQUE demonstrates precise makeup application for each prompt, regardless of the targeted region or the subject's demographics.

To evaluate color variation, we provide examples featuring both a male and a female subject. Additionally, by testing prompts that target different facial regions (eyes, lips, cheeks, skin, and full-face makeup), we demonstrate MASQUE's ability to generate realistic makeup across various regions. We further provide visualizations without $\mathcal{L}_{\text{edit}}$ optimization to illustrate its impact.

Table 10: DSR of `MASQUE` on CelebA-HQ and VGG-Face2-HQ against adversarial purification across four FR models under identification and verification mode.

| Dataset | Identification | | | | Verification | | | |
|---|---|---|---|---|---|---|---|---|
| | IR152 | IRSE50 | MobileFace | FaceNet | IR152 | IRSE50 | MobileFace | FaceNet |
| CelebA-HQ | 0.28 | 0.36 | 0.55 | 0.28 | 0.12 | 0.10 | 0.18 | 0.17 |
| VGGFace2-HQ | 0.88 | 0.80 | 0.83 | 0.77 | 0.86 | 0.82 | 0.92 | 0.78 |

Table 11: Full comparison results of `MASQUE`'s DSR under various image transformations.

| Method | Identification | | | | Verification | | | |
|---|---|---|---|---|---|---|---|---|
| | IR152 | IRSE50 | MobileFace | FaceNet | IR152 | IRSE50 | MobileFace | FaceNet |
| MASQUE | 1.00 | 0.97 | 0.94 | 0.71 | 0.97 | 0.95 | 0.69 | 0.66 |
| + Resize | 0.97 | 0.95 | 0.91 | 0.71 | 0.92 | 0.92 | 0.67 | 0.65 |
| + Compression | 1.00 | 0.95 | 0.92 | 0.72 | 0.96 | 0.93 | 0.65 | 0.63 |
| + Gaussian Noise | 0.98 | 0.95 | 0.95 | 0.82 | 0.96 | 0.92 | 0.90 | 0.81 |
| + Blur | 1.00 | 0.95 | 0.91 | 0.73 | 0.97 | 0.93 | 0.70 | 0.65 |
| + Fog | 0.95 | 0.97 | 0.93 | 0.88 | 0.92 | 0.97 | 0.84 | 0.81 |
| + Makeup Removal | 0.95 | 0.94 | 0.91 | 0.85 | 0.93 | 0.92 | 0.69 | 0.50 |
| + DDPM Denoising | 0.88 | 0.88 | 0.77 | 0.67 | 0.75 | 0.84 | 0.85 | 0.67 |

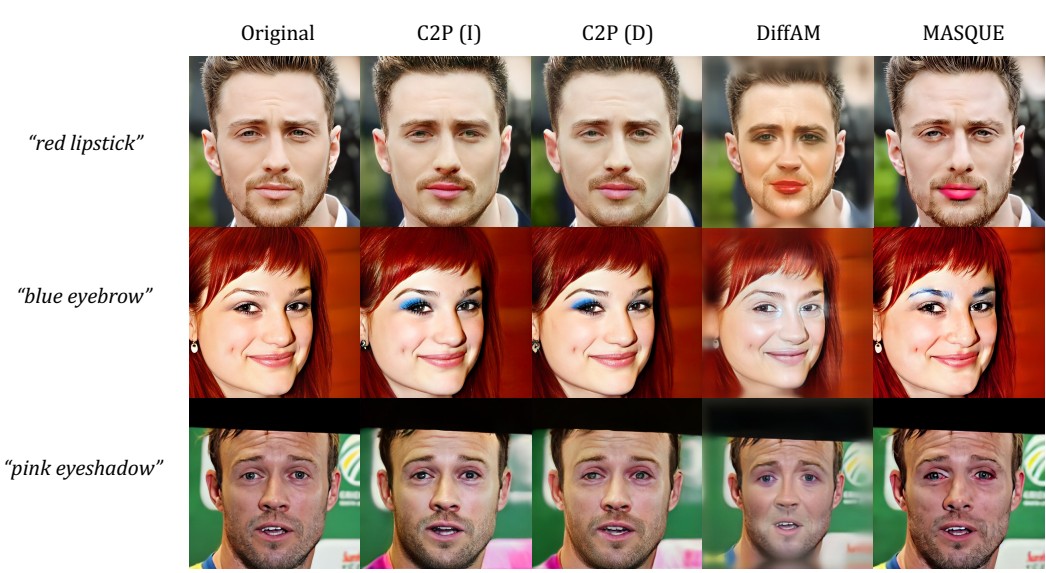

Figure 8: Visualizations of `MASQUE`-generated adversarial images from the VGG-Face2-HQ Dataset.

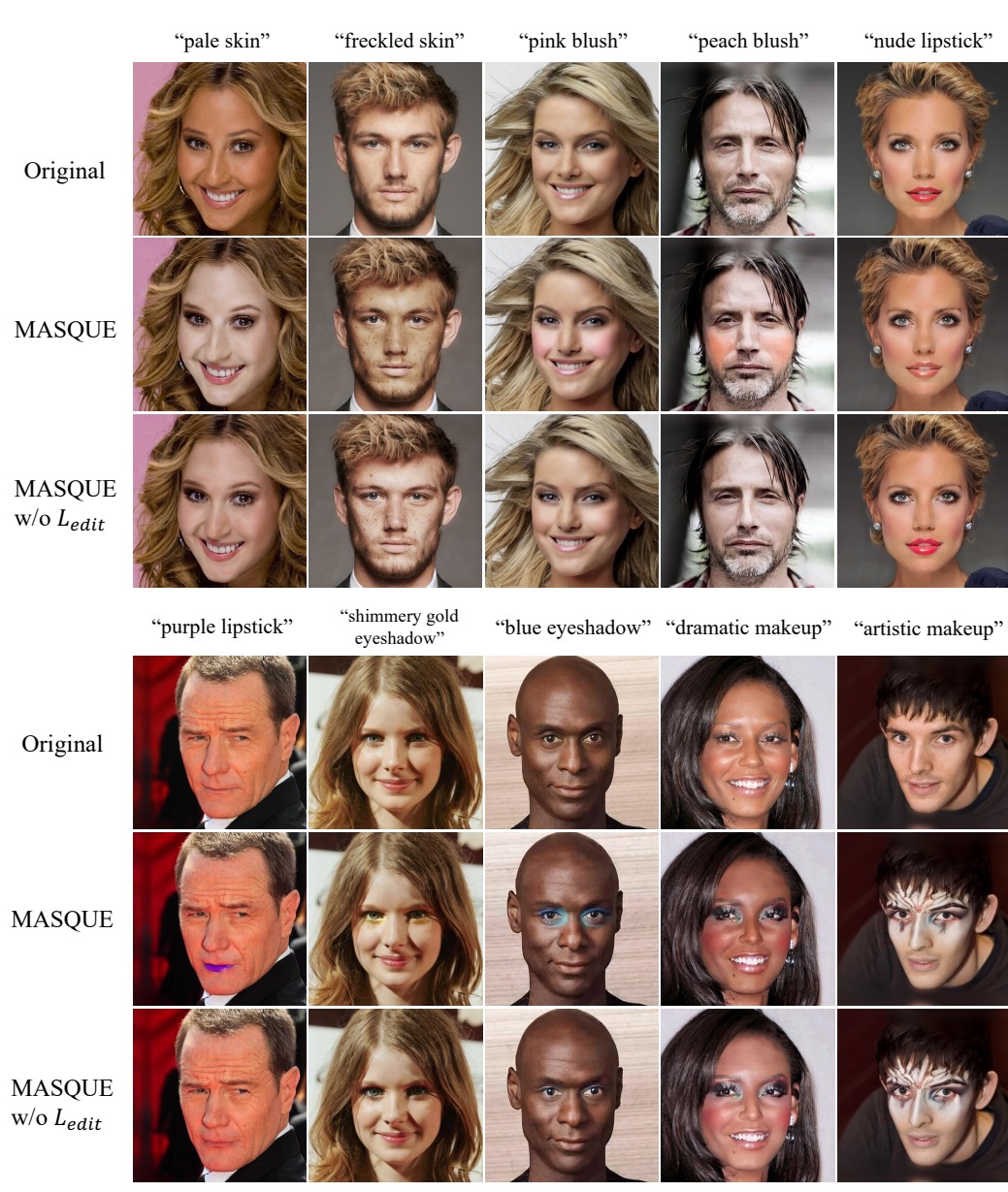

Figure 9: Visualizations of MASQUE-generated adversarial images with or without using the edit loss.

