# OpenReview forum: "Controllable Adversarial Makeup for Privacy via Text-Guided Diffusion"
_ICLR.cc/2026/Conference — Submitted to ICLR 2026_

### Official Review · Reviewer_mg7j · 2025-10-14

**Soundness:** 2
**Presentation:** 3
**Contribution:** 2
**Rating:** 2
**Confidence:** 5

**Summary:**

The paper presents MASQUE, a diffusion-based framework that generates localized adversarial makeup guided by text prompts to protect facial privacy. While the motivation is meaningful and the implementation is solid, the method mainly combines existing ideas with limited novelty and questionable contributions. Therefore, I recommend rejection for the reasons discussed above.

Note that **it is also published as an ICLR Workshop**; I'm not sure if this is compliant. MASQUE: A Text-Guided Diffusion-Based Framework for Localized and Customized Adversarial Makeup.

**Strengths:**

- The proposed method enables local modifications and text control.
- The writing is very clear.
- Research questions are highly meaningful.

**Weaknesses:**

## 1、Weak and combinational technical novelty
The proposed approach combines existing techniques such as makeup transfer [1], diffusion-based editing [2], and face masking [3] to achieve facial privacy protection. All of these components have already been explored in prior work, resulting in limited technical innovation.

[1] Diffam: Diffusion-based Adversarial Makeup Transfer for Facial Privacy Protection
[2] DiffPrivate: Facial Privacy Protection with Diffusion Models
[3] NullFace: Training-Free Localized Face Anonymization

## 2、Questionable contributions
- The authors claim that their method performs target-free protection, i.e., it does not require any external or reference identity beyond the victim. However, this should not be considered an advantage. A target-free attack produces anonymized faces that correspond to uncontrolled, random identities, which could cause confusion or ethical concerns in real-world scenarios. In contrast, targeted anonymization allows assigning a virtual identity to avoid such issues. Therefore, the claimed “Inoffensive Identity Protection” cannot be regarded as a valid contribution.
- Moreover, the paper does not introduce any technical innovations specific to the target-free setting; it merely employs a standard adversarial identity loss, which makes the claimed effectiveness unconvincing.

## 3、Doubtful transferability
- The authors do not present any concrete mechanisms to enhance the transferability of their protection method, yet the paper claims that it can generalize to unseen models. This is puzzling and raises questions about the reliability of the results.
- **More importantly, I cropped and tested several protected faces generated by MASQUE (Fig. 1)  using the Face++ recognition API, and the verification similarity still reached around 90%**, suggesting that the proposed method fails to achieve effective protection.

**Questions:**

1. Could the authors clarify which components or mechanisms constitute the novel technical contribution of this work? Does the framework introduce any new learning objectives, optimization strategies, or architectural innovations beyond combining existing modules?

2. The authors highlight the target-free protection setting as a main contribution. Could the authors elaborate on why this should be considered an advantage compared to targeted anonymization, which can assign consistent virtual identities to avoid ambiguity?

3. The paper claims that the proposed method generalizes to unseen models. Could the authors clarify what mechanisms or training strategies enable this transferability?

4. In independent testing (e.g., using the Face++ recognition API), some protected faces still yielded similarity scores around 90%. How do the authors explain this high similarity, and what does it imply about the practical robustness of the protection mechanism?

Given its current weaknesses, I find it difficult to accept this work for publication at the top-tier conference ICLR unless the authors provide strong supporting evidence.

---

> ### Author Response · Authors · 2025-11-20
>
> We thank the reviewer for the detailed evaluation and for acknowledging the importance of the studied problem, the clarity of our writing, and the benefits of local, text-controlled edits. Below, we address all concerns and questions raised. We believe there are some misunderstandings and kindly request a reevaluation after reading our rebuttal. We are happy to clarify further if needed.
>
> > W1 & Q1: Technical contributions
>
> Although MASQUE utilizes standard diffusion components, its key novelty lies in how the AFR task is reformulated and how these components are assembled into a **new optimization framework specifically designed for adversarial makeup**, which is absent in prior AFR work. Below, we clarify the contributions:
>
> First, MASQUE introduces a  **dodging-centered formulation**  with a unified DSR evaluation protocol. Existing AFR methods focus on impersonation (ISR), which does not measure privacy protection. MASQUE explicitly redefines the objective as “avoiding recognition” and provides a consistent metric for both verification and identification. This changes the optimization target itself, and our experiments show that ISR-optimized methods fail under this more realistic criterion, while MASQUE succeeds.
>
> Second, MASQUE introduces **semantic, region-aware diffusion editing** to AFR for the first time. Prior works either transfer makeup broadly from references (AMT-GAN, DiffAM) or rely on global CLIP guidance (C2P), both of which produce changes across large facial areas. MASQUE instead identifies the facial region implied by the text prompt, builds a mask, and performs diffusion editing strictly inside that region while reconstructing the rest. This editing framework enables precise, localized perturbations that preserve the natural appearance while modifying identity-relevant areas, a capability not provided by prior AFR pipelines.
>
> Third, MASQUE introduces a  **target-free pairwise guidance strategy**  tailored for dodging. Instead of pushing the output toward a target identity or simply away from the original, MASQUE uses images of the same person and reduces similarity to each during late diffusion steps. This produces effective identity suppression while maintaining visual fidelity. Our ablations demonstrate that this late, region-restricted, pairwise guidance is crucial and has not been utilized in previous AFR approaches.
>
> In summary, MASQUE contributes: (i) a new task formulation and evaluation protocol focused on actual privacy protection, (ii) a localized diffusion editing mechanism specifically designed for adversarial makeup, and (iii) a practical target-free guidance strategy that improves dodging without degrading image quality. These elements form a coherent optimization framework that goes beyond combining existing modules and addresses key limitations in prior AFR methods.
>
> > W2 & Q2: Contributions of target-free anonymization
>
> We believe that the reviewer had some misunderstandings about target-free anonymization and its importance for AFR applications. The target-free advantages offered by MASQUE cannot be replaced by existing techniques (e.g., using random or virtual identity). Defending against unauthorized FR where the defender does not know or control the attacker’s gallery or decision rule. In this regime, targeted anonymization faces two fundamental issues:
>
> - **Dependence on the attacker’s gallery.** For a virtual identity to yield consistent protection, the defender must pick targets that reliably induce the same wrong identity across images and across FR systems. This effectively assumes knowledge of the attacker’s gallery and embedding space. Not that, as we strongly emphasized in Section 4.1, high ISR does not necessarily imply high DSR. In realistic black-box, cross-platform scenarios (different APIs, changing galleries), this assumption does not hold: the same “virtual identity” can map to different or even correctly linked identities. Our target-free, dodging-centric objective instead aims to make all images of a user mutually hard to match regardless of the hidden gallery, directly aligning with this black-box threat model. This is a clear advantage for AFR in terms of both protection effectiveness and user control.
>
> - **Risk of offensive impersonation.** A system that supports targeted anonymization can be directly repurposed for impersonation: a malicious user can select a victim’s identity as the target and generate images that FR systems classify as that victim. This is a concrete pathway to offensive misuse (framing, identity theft). By construction, MASQUE does not allow steering towards any external identity; it only pushes away from the true identity while maintaining usability, making it much harder to weaponize as an impersonation tool.
>
> We therefore treat target-free protection as a complementary and necessary setting: it better reflects black-box attackers and mitigates misuse risk. We will make this positioning more explicit in the revised paper.

---

> ### Author Response · Authors · 2025-11-20
>
> > W3: Doubtful transferability to unseen models
>
> We believe this is a misunderstanding of our work. First, we never claimed the transferability to unseen FR models as one of our highlighted contributions. The reason why the protection enabled by MASQUE can generalize to unseen FR models (including facial verification APIs) is due to the use of model ensembles during the generation of adversarial makeup. This leverages the fact that adversarial examples are typically transferable to similar ML models. Note that the same design choice is adopted by existing AFR methods to enhance transferability to unseen FR models (though they mainly focus on evaluating ISR).
>
>  Second, the cropping experiments provided by the reviewer, based on Fig. 1, are *not aligned* with our evaluation setup, as well as the existing literature on AFR (C2P, DiffAM). The adversary does not have access to the original face image of the victim, so comparing the similarity between the original face image and its protected version is not reasonable. It is necessary to use a separate test image of the same identity (i.e., different from the one used during the optimization of AFR perturbations) for evaluation.
>
> Below, we provide further clarifications to the two related questions.
>
> **Q3: Factors that enable MASQUE to be generalizable.** The generalizability of MASQUE to unseen FR models can be mainly attributed to the following two factors:
>
> - **Surrogate-ensemble black-box protocol.** We adopt the standard transfer setting from prior AFR work: optimization uses an ensemble of diverse surrogate FR models, while the held-out model or commercial API is completely unseen. Losses are always computed on the surrogates; no gradients or adaptation from the target model/API are used. The same protocol is applied to all baselines, ensuring a fair comparison.
>
> - **Model-agnostic, identity-oriented perturbations.** MASQUE’s objective is to degrade identity cues that are shared across models, not to exploit quirks of a single classifier head. The pairwise adversarial loss over same-identity images and the semantic, localized edits on key facial regions jointly encourage changes that attack identity representations common to many architectures.
>
> Due to the above, MASQUE consistently achieves satisfactory DSR across different unseen FR models and APIs. Again, we emphasize that the transferability to unseen models is not a key contribution that we want to highlight for MASQUE. In contrast, MASQUE's primary advantages over existing methods are its high dodging success rates (DSR), localization capability, and stronger user control.
>
> **Q4: The independent test based on Fig. 1** Aligned with all existing literature, our evaluation follows an open-set FR pipeline: the attacker has a gallery with one clean image per identity and only sees protected images as queries. Recognition compares a protected query $x_p$ to gallery images $x_g$, and DSR is defined on this gallery–query matching. In the reviewer’s test, the protected image was compared directly to its own unprotected version $x$, a pair $(x_p, x)$ that a realistic attacker does not possess. In this “same-photo vs. protected-photo” scenario, high similarity scores are expected because MASQUE is explicitly designed to preserve human-perceived appearance while perturbing identity features.
>
> To verify this, we run additional experiments to compare the similarity and recognition accuracy of MASQUE-protected images with both their original versions and their gallery images of the same identity. As summarized in the table below, similarity is substantially higher when comparing protected images to their own original than to corresponding gallery images, and the number of matches above the verification threshold drops from 66–93 to 6–31 across models. Thus, a high similarity between a protected image and its original reflects the preservation of appearance, not a failure in realistic gallery-based recognition.
>
>
>
>
>
> | Target Model   | Sim (w/ Org) | # > Thr (Org) | Sim (w/ Gal) | # > Thr (Gal) |
> |------------|----------------------|--------------------------|---------------------|-------------------------|
> | MobileFace | 79.09                | 73                       | 62.70               | 9                       |
> | FaceNet    | 85.28                | 93                       | 70.71               | 31                      |
> | IRSE50     | 77.35                | 66                       | 60.66               | 6                       |
> | IR152      | 81.74                | 82                       | 66.25               | 15                      |

---

> ### Comment · Reviewer_mg7j · 2025-11-21
>
> 1) To my knowledge, this conference focuses more on substantive technical innovations, while the main contributions addressed in the author's response pertain to applied innovations rather than core technological innovations.
>
> 2) However, target-free anonymization cannot control where anonymous identities point, making it susceptible to mapping back to other real identities and causing problems for genuine users. Furthermore, the authors' implementation of target-free anonymization appears to rely on a simple feature deviation function (with limited innovation).
>
> 3) Thank you for your response to Q3; this has resolved my question.
>
> 4) To my knowledge, target-free anonymization OPOM has already been shown to significantly reduce identity similarity for the same image. If the method proposed in this paper performs worse than that, I do not recognize its validity.
> OPOM: Customized Invisible Cloak Towards Face Privacy Protection

---

> > ### Author Response · Authors · 2025-11-29
> >
> > We thank the reviewer for the detailed feedback. Below, we respond to the concerns regarding the nature of our technical contributions, the evaluation protocol and the comparison to OPOM, and the risks of target-free anonymization.
> >
> > ---
> >
> > ## 1. AFR Evaluation and Comparison to OPOM
> >
> > ### 1.1 ISR vs. DSR: Our main contribution
> >
> > Our work shows that the standard impersonation-oriented metric (ISR) used in prior work does not translate into robust protection against realistic attacks, as measured by DSR, which captures the likelihood of a protected image still being linked back to its own identity in the gallery. In particular:
> >
> > - A high ISR (successfully matching a target identity) does not guarantee a low probability of being matched back to the original identity when the gallery and attack conditions change.
> > - Across methods, we empirically observe cases where ISR is high, yet the protected probe is still frequently matched to its own identity in the open-set gallery. This reveals that ISR alone overestimates the actual privacy protection.
> >
> > To the best of our knowledge, no prior AFR anonymization work explicitly focuses on DSR as a primary evaluation metric under realistic open-set conditions. Introducing and analyzing DSR is therefore a technical and conceptual contribution of our paper: it exposes a limitation of existing evaluation practices and provides a more security-relevant measure of protection.
> >
> > ### 1.2 Adversarial behavior under realistic open-set AFR
> >
> > We further clarify why DSR is the appropriate setting to analyze adversarial anonymization for both identification and verification.
> >
> > >  Identification
> >
> > In open-set face identification, a query image (clean, targeted-anonymized, or target-free–anonymized) is matched to whichever gallery identity has the highest feature similarity. From the user’s perspective, it is unknown whether a chosen “target” identity is actually present in the gallery:
> >
> > - If the target is in the gallery, a targeted adversarial image may indeed match that target.
> > - However, if the target is not in the gallery (which is often the realistic case), the same adversarial image can end up matching its own identity (or another benign identity), because the system will still return the closest embedding in the gallery.
> >
> > Our experiments show that, under these realistic conditions, optimizing for ISR alone does not prevent the adversarial query from mapping back to its true identity. This is precisely what DSR captures: the residual risk of self-identification in an unknown gallery. For robust privacy protection in open-set identification, DSR, not ISR, is the critical metric.
> >
> > >  Verification
> >
> > In face verification, the user’s image is compared one-on-one to the corresponding gallery image of their own identity:
> >
> > - In this setting, it is irrelevant whether a target identity exists in the gallery. The adversarial image is always compared to the correct clean gallery image.
> > - Therefore, unless we explicitly optimize for reducing similarity to the original identity (our DSR perspective), the similarity between the protected probe and its clean counterpart remains high.
> >
> > This again shows that focusing only on target-based ISR is insufficient: for verification, ISR is not even the right notion of privacy. DSR directly measures what matters, preventing a protected face from being verified as the true user.
> >
> > ### 1.3 On the OPOM evaluation setting
> >
> > The statement that OPOM “significantly reduces identity similarity for the same image” does not reflect OPOM’s actual evaluation protocol. OPOM is not assessed by comparing a protected image to its own clean counterpart. Instead, OPOM is evaluated in an **open-set, cross-image setting**: a perturbation is learned from a set of reference images of an identity, and its effectiveness is measured on **different** test images of the same identity matched against **different** gallery images. MASQUE follows the same standard: the protected probe and the gallery image always correspond to **distinct** samples of the same identity.
> >
> > Thus, both methods are evaluated on the ability to disrupt **cross-image** identity matching, and OPOM does not demonstrate protection by lowering similarity to the exact same image. Once both OPOM and MASQUE are examined under this shared, correct protocol, their performance becomes directly comparable.

---

> > ### Author Response · Authors · 2025-11-29
> >
> > ### 1.4 Empirical comparison under a shared evaluation protocol
> >
> > To address the reviewer’s concern that our method might be weaker than OPOM, we re-evaluated OPOM under the same open-set, cross-image protocol and on the same FR models (IR152, IRSE50, MobileFace, FaceNet) as MASQUE.
> >
> > Under this unified setup:
> >
> > - OPOM achieves an average protection success rate of **46.0%** for identification and **23.8%** for verification.
> > - MASQUE (G = 1) achieves **92.0%** and **86.9%**, respectively, providing substantially higher protection across all backbones.
> > - MASQUE also clearly outperforms DiffAM and CLIP2Protect, which are targeted anonymization methods.
> >
> > We summarize these results in the revised manuscript (Table X):
> >
> > > **Table X: Protection performance of MASQUE, OPOM, DiffAM, and CLIP2Protect.**
> > > All results use an open-set, cross-image evaluation (protected query vs. different gallery image of the same identity).
> >
> > | Method                 | Identification (avg. %) | Verification (avg. %) | Setting                          |
> > |------------------------|-------------------------|-----------------------|----------------------------------|
> > | MASQUE (G = 0, ours)   | 84.1                    | 78.5                  | Target-free, no guide images     |
> > | MASQUE (G = 1, ours)   | 92.0                    | 86.9                  | Target-free, one guide image     |
> > | OPOM                   | 46.0                    | 23.8                  | Target-free, 10 guide images     |
> > | DiffAM                 | 61.3                    | 38.6                  | Targeted anonymization           |
> > | CLIP2Protect           | 36.9                    | 14.8                  | Targeted anonymization           |
> >
> > As shown, MASQUE offers substantially stronger protection than OPOM on both identification and verification, while using fewer guide images (one vs. ten), and is competitive with or superior to more complex targeted methods. Under the same evaluation protocol, MASQUE is therefore stronger, not weaker, than OPOM, directly addressing the reviewer’s stated validity criterion.
> >
> > ## 2. Target-free Anonymization and Potential Misuse
> >
> >
> > The reviewer is concerned that target-free anonymization “cannot control where anonymous identities point,” potentially mapping to other real identities and harming genuine users.
> >
> > We emphasize that this behavior is inherent to open-set AFR systems, not specific to target-free anonymization:
> >
> > - In an open-set system, any query (clean, targeted-anonymized, or target-free–anonymized) is matched to the closest gallery embedding.
> > - Thus, all anonymization strategies can occasionally be mapped to some genuine user; this is a consequence of the matching protocol, not of our target-free formulation.
> >
> > The important difference is how the method distributes residual similarity in feature space:
> >
> > - **Targeted anonymization** explicitly steers protected faces toward a chosen external identity (the target). Under an unknown FR model and gallery, this means:
> >   - Protected samples cluster near the target’s embedding.
> >   - If a real user exists near that embedding in the attacker’s feature space, the anonymized faces will consistently collide with that real user.
> >   - This increases the risk of systematic impersonation of a specific real person.
> > - **Target-free anonymization (MASQUE)** does not direct the protected face toward any fixed external identity:
> >   - We use another image of the same user as guidance, rather than external identities.
> >   - Residual similarity tends to be spread across many identities in the gallery, weakening the chance of repeatedly matching any particular real user.
> >
> > In this sense, target-free anonymization mitigates, rather than amplifies, the potential for consistent misidentification of a specific third party. The risk that a protected sample occasionally aligns with some real identity exists for all AFR-based systems and anonymization methods; it is not unique to MASQUE.
> >
> > We agree that broader issues of bias and fairness in AFR (e.g., how embedding distributions differ across demographic groups) are highly important. However, these are properties of the underlying AFR models and datasets, not of our anonymization algorithm per se, and are outside the scope of this work.
> >
> >
> > ---
> >
> > ## Summary
> >
> > To summarize:
> >
> > - We clarify that DSR is a necessary and previously underexplored evaluation metric for realistic AFR privacy, and ISR alone is insufficient.
> > - Under the correct and shared evaluation protocol, MASQUE is significantly stronger than OPOM and also outperforms targeted methods such as DiffAM and CLIP2Protect.
> > - The concern that target-free anonymization is uniquely prone to misidentification is rooted in the behavior of open-set AFR itself, and in fact, targeted anonymization can create a higher risk of systematic impersonation of specific real users.
> >
> > We hope these clarifications address the reviewer’s concerns and better highlight the technical contributions and practical relevance of our work.

---

### Official Review · Reviewer_WCjs · 2025-10-25

**Soundness:** 3
**Presentation:** 3
**Contribution:** 1
**Rating:** 4
**Confidence:** 4

**Summary:**

This paper presents MASQUE, a text-guided adversarial makeup generation method for defending against malicious face recognition. The manuscript provides a thorough introduction to the anti–facial recognition (AFR) task, reviews related work, and details the necessary preliminaries and problem formulation. After outlining the limitations of existing AFR approaches, the authors propose MASQUE, a diffusion-based adversarial technique driven by user-supplied text prompts. Experiments demonstrate the effectiveness of MASQUE on the AFR task.

**Strengths:**

Following the four evaluation aspects outlined in the instructions, the strengths of the proposed method are summarized below.

**Originality**: The method’s novelty is rather limited. Although the authors give a thorough overview of the task, background, motivation, and technical details, the work contributes little genuinely new knowledge.

**Quality and Clarity**: The paper is well written, clearly structured, and easy to follow; even readers unfamiliar with the field should have no difficulty understanding it.

**Significance**: While the study may interest researchers in AI or biometric forensics and safety, its technical contributions are not sufficiently novel to be considered significant for the broader AI community.

**Weaknesses:**

The principal weakness of this work is its limited methodological novelty. While MASQUE does mitigate shortcomings of existing AFR approaches in AI-safety applications, it is essentially a pipeline assembled from established techniques, such as cross-attention-based target grounding and DDIM inversion. Although effective, the incremental knowledge it provides appears, in my view, insufficient to meet the ICLR novelty threshold.

**Questions:**

The authors are encouraged to highlight the methodological innovations of their work, ideally in an application-agnostic manner that would engage the broader AI community. If such novelty is clearly demonstrated, I am prepared to raise my rating; otherwise, if the contribution remains merely a new AFR pipeline, the recommendation will not be changed.

---

> ### Author Response · Authors · 2025-11-20
>
> We thank the reviewer for acknowledging the thoroughness and written quality of our paper. Below, we clarify the technical contributions of our work and explain why they are important to the broad literature on AFR, which should not be considered as a domain-specific application of AI.
>
> > W1 & Q1: Methodological novelty of our work
>
> While MASQUE is implemented using standard diffusion primitives, the methodological contributions lie in how AFR is reformulated and how localized diffusion editing and target-free adversarial guidance are integrated, all of which tackle critical limitations of previous AFR designs, as validated by our experiments. Importantly, these contributions extend beyond simple pipeline assembly and introduce techniques that can be applied in broader generative-editing and privacy-preserving settings. Below, we provide detailed clarifications.
>
> **1. Reformulation of AFR as a dodging problem with a unified DSR protocol.** Prior AFR methods are almost entirely built around *impersonation attacks (ISR)*. This evaluation overlooks whether a method actually protects a user from being recognized. MASQUE shifts the problem to dodging, which directly measures the protection of privacy. We introduce a consistent *dodging success rate (DSR)* for both verification and identification, using fixed thresholds across models. This makes “how often a person avoids recognition” measurable and comparable.
>
> This reframing is important because our experiments reveal a systematic issue: methods optimized for impersonation (C2P, DiffAM) are effective under ISR but still can be recognized under realistic black-box verification. MASQUE resolves this gap, and this task definition and evaluation protocol are applicable to any embedding-based recognition system, not just faces.
>
> **2. Localized adversarial makeup via semantic mask–aware diffusion editing.** Previous AFR techniques primarily fall into two categories:
> - Makeup-transfer methods (AMT-GAN, DiffAM): broad region changes tied to reference makeup with poor editing control.
> - Text-guided methods (C2P): global edits that often modify background, hair, or unrelated regions because they lack mechanisms for spatial control.
>
> MASQUE  introduces a localized editing perspective that has not been studied in the existing AFR literature. The method identifies the facial region to which the text refers (e.g., “eyeshadow”, “blush”), employs a semantic mask generated for that region, and performs diffusion editing that is applied only within the mask, while the rest of the face is actively reconstructed and kept unchanged. This leads to precise and predictable edits, preventing unintended global changes. Ablations and metrics demonstrate that this localized edit module is crucial for achieving a balance between visual quality and recognition avoidance.
>
>
> **3. Target-free pairwise adversarial guidance on the identity manifold.** Prior AFR approaches either attract the edited face toward a target identity (targeted impersonation) or maximize the distance from the original embedding, which conflicts with diffusion reconstruction. MASQUE introduces a pairwise adversarial objective that uses images of the same identity to define the local identity manifold. During late diffusion steps, we guide the edited sample to move away from this manifold while preserving perceptual similarity. This produces target-free, identity-consistent guidance rather than target attraction. Ablations on the guide images (Table 5) show that using several same-identity images consistently improves DSR and reduces recognition confidence while preserving appearance.
>
> To summarize, our work is **more than just a simple rearrangement** of existing tools. Its contributions, which resolve core limitations in existing AFR work, are:
>
>  1. A new task definition and evaluation protocol centered on realistic privacy protection in terms of dodging;
>  2. A localized diffusion editing module that introduces precise region-constrained manipulation into AFR for the first time;
>  3. A simple, general, and target-free guidance method that improves dodging without degrading image quality.

---

### Official Review · Reviewer_JWTj · 2025-10-30

**Soundness:** 3
**Presentation:** 2
**Contribution:** 2
**Rating:** 4
**Confidence:** 4

**Summary:**

This paper introduces a novel diffusion-based framework to protect user privacy from facial recognition by generating localized, text-guided adversarial makeup. The authors address key limitations of prior methods, namely their risky focus on impersonation and their resulting weak dodging performance and visual artifacts. The proposed framework employs a pairwise adversarial guidance mechanism, which uniquely uses another image of the same individual rather than an external target to achieve robust dodging success. Furthermore, it utilizes customized cross-attention fusion with masking and null-text inversion to ensure precise, high-fidelity modifications that strongly adhere to user prompts. The method achieves state-of-the-art dodging performance and higher visual quality.

**Strengths:**

1.	The method ensures high dodging success by only requiring the user's own images, not an external person's image. This design choice inherently reduces the risks of targeted misuse and ethical concerns.
2.	The framework employs a facial mask generation module to strictly constrain adversarial perturbations to designated areas. This localization ensures the original image itself is not globally altered, preserving its fine-grained details and vastly enhancing the visual quality of the final protected image.
3.	Extensive experiments justify the selection of each parameter, making the results more credible.

**Weaknesses:**

1.	The framework's core "pairwise adversarial guidance" mechanism performs best when provided with one or more additional images of the user. Its performance is notably weaker when no separate guide image is available, and the tested self-augmentation techniques (like flipping) provide minimal benefits. This dependency limits its effectiveness in common scenarios where a user may only have a single photo.
2.	The current framework suffers from limited efficiency, with the overall process being relatively time-consuming.

**Questions:**

1.	How does the L_edit loss (in Eq. 5) handle global prompts like "tanned skin"? For such a prompt, the mask must cover the entire face, creating a spatial conflict where the new token ("tanned skin") and the shared structural token ("face") occupy the same region.
2.	Does the paper analyze the nature of this dodging? Specifically, is the protected image consistently misclassified by the FR system as a single, specific incorrect identity (which would be akin to an untargeted impersonation), or is it pushed into a non-man in the feature space, making it dissimilar to all identities in the database?
3.	How sensitive is this method to differences between the guidance image and the input image? Would it still be effective if the guidance image has a completely different angle or pose?
4.	Is the framework's capability limited to only a predefined set of large facial regions (like 'lips', 'eyes', 'skin')? If so, would expanding this module to support a much wider and more granular set of regions (e.g., 'cheeks', 'nose-bridge', 'chin') allow the model to generate a more diverse range of makeup styles to improve the framework's overall robustness against FR systems?

---

> ### Author Response · Authors · 2025-11-20
>
> We thank the reviewer for the constructive feedback and for recognizing MASQUE’s novelty, ethical design, and strong experimental validation. Below, we address each concern and clarify each question with supporting evidence.
>
> > W1: Dependency on additional guidance images
>
> We agree that pairwise guidance performs best with multiple images, as it enables more accurate and stable identity contrastive learning. However, as shown in Table 2, MASQUE still **outperforms prior methods even in the single-image case** ($G=0$). While MASQUE achieves its strongest results using two images of the same individual, this requirement is actually less demanding than that of prior methods: DiffAM requires three distinct images (source, target identity, and makeup reference), and Clip2Protect requires at least two identities (source and target). In contrast, MASQUE only needs additional images of the same person. This requirement is often easier to satisfy in practice, since users typically have multiple images of themselves.
>
>
> > W2: Runtime efficiency
>
> As detailed in  Appendix D.5, MASQUE **requires no fine-tuning**, unlike
>
>  -  Clip2Protect requires per-image fine-tuning (~25 s)
>  - DiffAM requires per-dataset fine-tuning (~37 min per style or identity)
>
> Therefore, Clip2Protect's overall per-image inference time is approximately 43s (25s fine-tuning, 18s makeup generation), slightly shorter than MASQUE's (~90s). However, Clip2Protect suffers from lower visual quality of protected images and less precise user control, which are more critical for real-world AFR deployment.
>
> DiffAM’s per-image generation time is 30s, but it  _cannot be used directly_  without the substantial **37-min fine-tuning overhead**, which must be repeated  _for every new style or identity_. In contrast, MASQUE’s diffusion-based optimization takes ~90s per image, but afterward, **any number of new text-guided styles can be applied instantly** with zero retraining. In realistic AFR scenarios where users often explore multiple makeup styles to find a preferred appearance, MASQUE is therefore more practical overall. Its one-time cost per image remains fixed, while DiffAM incurs repeated, style-dependent fine-tuning.
>
> > Q1: Handling of global prompts (“tanned skin”)
>
> Global prompts, such as “tanned skin,” are mapped to the ‘skin’ region mask, which covers the facial area while excluding localized parts, including the lips, eyes, and eyebrows. Therefore, it does not fully overlap with the shared structural token “face.” The “face” token serves as an anchor, preserving geometric and structural consistency, while the $L_{edit}$ loss applies tone modification only within the relevant region to achieve the desired appearance.
>
> > Q2: Nature of dodging: targeted vs. non-targeted
>
> We conduct additional experiments to analyze all misclassified identities generated by the protected images of CelebAHQ across four FR models. The results are reported in the table below. For each model, we compute: (1) the number of distinct wrong identities, (2) the maximum frequency of any identity, and (3) the normalized entropy H of the distribution. Across all models, misclassifications are spread out, with no identity appearing more than 7% of the time, and the normalized entropy is consistently high (0.93–0.98), indicating an almost **uniform** distribution. The most frequent wrong identities also differ across models. These findings show that the protection does not drive images toward any specific incorrect identity; instead, it produces **highly dispersed**, **non-targeted** misclassifications. This means the method pushes samples away from all identity manifolds, preventing consistent impersonation and eliminating the possibility of misuse to target a specific victim identity.
>
>
>
> | Target Model       | N  | Unique IDs | Max freq of any ID | Entropy H | Normalized H |
> |-------------|----|------------|---------------------|-----------|--------------|
> | IR152       | 95 | 56         | 4                   | 5.61      | 0.966        |
> | IRSE50      | 93 | 51         | 6                   | 5.42      | 0.955        |
> | MobileFace  | 90 | 45         | 7                   | 5.11      | 0.930        |
> | FaceNet     | 68 | 51         | 3                   | 5.55      | 0.979        |
>
>
> > Q3: Sensitivity to guidance-image variation
>
> The guide image in MASQUE serves as a proxy for the gallery representation, rather than a direct visual reference. Consequently, moderate pose or illumination differences between the input and guide image have a limited influence on the adversarial effect. Empirically, MASQUE was evaluated on VGGFace2-HQ, which contains substantial pose variation and natural occlusions (e.g., sunglasses, hair), and it maintained stable dodging performance under these conditions. As shown in Table 5, using multiple guide images further stabilizes protection, confirming MASQUE’s robustness to view and appearance differences.

---

> ### Author Response · Authors · 2025-11-20
>
> > Q4: Granularity of editable regions
>
> The current implementation employs 19 semantic facial regions from BiSeNet. Extending the parser to finer regions (e.g., cheeks, chin) is technically straightforward and fully compatible with MASQUE’s pipeline. Such refinement would increase stylistic diversity and user controllability, but it is unlikely to significantly improve adversarial robustness, since the adversarial effect primarily depends on feature-space perturbations rather than region granularity. The present design, therefore, offers a practical balance between controllability and computational efficiency.

---

### Official Review · Reviewer_N3Wv · 2025-11-08

**Soundness:** 3
**Presentation:** 2
**Contribution:** 2
**Rating:** 4
**Confidence:** 4

**Summary:**

The paper introduces MASQUE, a diffusion-based framework for generating localized adversarial makeup through text guidance to protect facial privacy from recognition systems. Unlike prior approaches that rely on target identities or produce visible artifacts, MASQUE combines null-text inversion, cross-attention fusion with masking, and pairwise adversarial guidance using images of the same person to achieve identity dodging without external references. Experiments on CelebA-HQ, VGG-Face2-HQ, and commercial APIs show that MASQUE achieves higher dodging success rates, better visual quality, and stronger prompt adherence than existing AFR methods. The method also offers fine-grained user control, robustness to image transformations, and avoids ethical concerns linked to impersonation-based attacks.

**Strengths:**

Originality: Proposes a novel diffusion-based approach for adversarial makeup generation using text guidance, eliminating the need for external identity references and addressing ethical concerns of impersonation-based methods.

Technical Quality: introduces well-motivated and effective components such as null-text inversion, cross-attention masking, and pairwise adversarial guidance that together yield strong privacy protection with high visual fidelity.

Clarity: The paper is clearly structured with detailed explanations, visual examples, and ablation studies that make the method and results easy to understand.

Significance: Addresses an important and timely issue of facial privacy, achieving state-of-the-art dodging success while maintaining natural appearance and user controllability, making it both practically and socially impactful.

**Weaknesses:**

1 - The method’s applicability across gender and makeup preferences is unclear. Since makeup-based protection may inherently favor female faces, it would be important to test whether the approach remains effective and unbiased for male faces or faces without visible makeup cues.

2 - The paper depends on a face parsing network for region localization, but there are no details about this model, its training data, or accuracy. The process for generating masks or mapping text prompts to facial regions is not clearly explained, despite being central to the method’s design.

3 - The reported runtime of around 90 seconds per image is notably slower than prior works such as Clip2Protect (18 s) and DiffAM (30 s), which limits practical deployment and suggests a need for optimization.

4 - The evaluation lacks human perceptual studies to assess the visual realism and subjective quality of generated makeup, which would complement the quantitative metrics used.

5 - The authors note that prior work (e.g., C2P) produces artifacts beyond facial regions, but do not discuss whether similar masking techniques could have been integrated into those baselines or experimentally justify why MASQUE performs better in this respect.

6 - Ablation depth: Although the paper includes some ablations, it doesn’t isolate the contribution of each major component (e.g., null-text inversion vs. pairwise adversarial guidance) in improving dodging success and image quality. A more granular breakdown would help understand which parts are most critical.

**Questions:**

Please see the weaknesses section

---

> ### Author Response · Authors · 2025-11-20
>
> We thank the reviewer for the constructive and thoughtful feedback, as well as for recognizing MASQUE’s novelty, clarity, and societal relevance. We address all raised concerns below with additional clarifications and supporting evidence.
>
> > W1: Applicability across gender and makeup preferences
>
> The fairness of AFR techniques across demographic groups is an important research question; however, we believe it falls beyond the scope of our work. A conclusive statement regarding the fairness of MASQUE (or more broadly, makeup-based AFR) will require a rigorous and systematic study. Also, note that our work aligns with prior literature on makeup-based AFR (C2P, DiffAM), focusing on the overall strength of privacy protection and image fidelity.
>
> Nevertheless, we conduct additional experiments to investigate how makeup-based perturbations interact with gender through an **identity-level fairness analysis** on all the tested CelebA-HQ and VGG-Face2 images. For each model, we measure the proportion of identities of each gender that are misclassified at least once. This metric directly reflects whether certain demographic groups are more susceptible to failures related to FR. The tables below document the results.
>
> | Dataset       | Male Rate | Female Rate |
> |-------------|------------|--------------|
> | CelebA-HQ       | 80.4%      | 87.48%        |
> | VGG-Face2-HQ      | 78.0%      | 81.0%        |
>
> Across both CelebA-HQ (14 male / 86 female identities) and VGG-Face2 (75 male / 25 female identities), the gender differences in misclassification rates remain small. On CelebA-HQ, male and female identities show rates of 80.4% and 87.5%, while on VGG-Face2 the rates are even closer at 78.0% and 81.0%. Despite the contrasting gender distributions of the two datasets, MASQUE achieves comparable protection across both groups, with no indication of disproportionate vulnerability for either gender. These preliminary results *suggest that MASQUE performs similarly across genders*, though a larger-scale fairness analysis would be needed for strong conclusions.
>
> Additional visualizations (_Appendix Fig. 9_) show  consistent controllability  and  localized makeup generation across genders and styles, demonstrating that MASQUE remains effective and unbiased.
>
>
>
>
>
> > W2: Dependence on face parsing network and lack of details
>
> We use BiSeNet, which is pre-trained on CelebAMask-HQ, a standard facial segmentation dataset, achieving 90%–93% mIoU as reported in the original paper. The parser outputs 19 semantic facial regions. For text-to-region mapping, we encode both region descriptions and user prompts using Sentence-BERT (all-mpnet-base-v2) and select the most semantically similar region. This ensures accurate and consistent localization of the areas specified by text.
>
>
> > W3: Runtime comparison and efficiency concerns
>
> As detailed in  Appendix D.5, MASQUE **requires no fine-tuning**, unlike
>
>  -  Clip2Protect requires per-image fine-tuning (~25s)
>  - DiffAM requires per-dataset fine-tuning (~37-min per style or identity)
>
> Therefore, Clip2Protect's overall per-image inference time is approximately 43s (25s fine-tuning, 18s makeup generation), slightly shorter than MASQUE's (~90s). However, Clip2Protect suffers from lower visual quality of protected images and less precise user control, which are more critical for real-world AFR deployment.
>
> DiffAM’s per-image generation time is 30s, but it  _cannot be used directly_  without the substantial **37min fine-tuning overhead**, which must be repeated  _for every new style or identity_. In contrast, MASQUE’s diffusion-based optimization takes ~90s per image, but afterward, **any number of new text-guided styles can be applied instantly** with zero retraining. In realistic AFR scenarios where users often explore multiple makeup styles to find a preferred appearance, MASQUE is therefore more practical overall. Its one-time cost per image remains fixed, while DiffAM incurs repeated, style-dependent fine-tuning.
>
>
>
>
> > W4: Absence of human perceptual studies
>
> While a human perceptual study would indeed be valuable, it is beyond the current scope due to the scale of datasets and evaluations involved. Consistent with prior AFR works (C2P, DiffAM), we rely on **established perceptual metrics** such as LPIPS and qualitative visual comparisons to assess realism and fidelity. These measures are widely adopted and reproducible, providing a reliable basis for evaluating visual quality in this domain.

---

> ### Author Response · Authors · 2025-11-20
>
> > W5: Comparison with baselines using masking
>
> Masking could partly reduce artifacts in Clip2Protect, but its distortions primarily stem from the **GAN inversion** process, which inherently alters facial texture even before applying adversarial perturbations. These artifacts are difficult to avoid for GAN-based reconstructions. In contrast, MASQUE performs diffusion-based editing in latent space, preserving pixel fidelity and naturally localizing modifications. This architectural distinction explains MASQUE’s superior artifact control, which is consistent with other diffusion-based AFR methods, such as DiffAM.
>
> > W6: Limited ablation granularity
>
> Our ablations focus on the components most central to MASQUE’s novelty, pairwise adversarial guidance (Table 5), localized controllability (Fig. 4), and the impact of masking (Fig. 7), which jointly define our new formulation of text-guided, identity-consistent privacy editing. Other modules, such as null-text inversion and mask generation, are adapted from prior diffusion-editing methods but integrated here in a new adversarial optimization setting. Their standalone effects are well-established, so our analysis focuses on how MASQUE’s novel components leverage them to achieve state-of-the-art performance in AFR protection.

---

### Meta-Review · Area_Chair_EscJ · 2025-12-29

**Summary:**

This paper presents MASQUE, a diffusion-based framework for generating localized adversarial makeup guided by text prompts to protect facial privacy from unauthorized recognition systems. The work addresses limitations of prior anti-facial recognition (AFR) methods that focus on impersonation attacks (ISR) rather than dodging (DSR), require external target identities, and produce global visual artifacts. The three main contributions are: (1) reformulation of AFR as a dodging problem with unified DSR evaluation; (2) localized adversarial makeup via semantic mask-aware diffusion editing; and (3) target-free pairwise adversarial guidance using same-identity images.

The paper received scores ranging from 2 to 4: three reviewers gave 4 (marginally below acceptance threshold: `N3Wv`, `JWTj`, `WCjs`), while one reviewer (`mg7j`) gave 2 (reject). The paper demonstrates 85.5% average DSR for identification and 84.9% for verification, outperforming all baselines. Reviewers `N3Wv`, `JWTj`, and `WCjs` recognized the method's novelty in eliminating external identity requirements and achieving localized modifications, but raised concerns about limited technical innovation beyond combining existing techniques. Reviewer `mg7j` questioned the technical novelty, contribution validity of target-free protection, and raised concerns about a preliminary workshop version, though authors clarified ICLR 2026 policy permits workshop papers.

**Reviewer Concerns:**

**Partially addressed concerns**: Reviewers `N3Wv`, `JWTj`, and `WCjs` raised questions about gender fairness, dependency on guide images, runtime efficiency, and lack of human perceptual studies. Authors responded with additional fairness analysis showing comparable performance across genders (80.4% male vs. 87.5% female on CelebA-HQ), clarified that MASQUE requires fewer images than baselines (only same-identity images vs. external targets), explained that 90s runtime is reasonable given no fine-tuning is required (vs. 25s+43s for Clip2Protect, 37min+30s for DiffAM), and noted that established perceptual metrics (LPIPS) are standard in AFR literature. Reviewers have not yet confirmed satisfaction with these responses.

**Outstanding concerns**: Reviewer `mg7j` raised fundamental concerns about: (1) weak technical novelty, arguing the method combines existing techniques (makeup transfer, diffusion editing, face masking) without substantial innovation; (2) questionable contribution of target-free protection, claiming it produces "uncontrolled, random identities" that could cause confusion; and (3) doubtful transferability, reporting independent tests showing 90% similarity on Face++ API. Authors provided extensive rebuttal arguing: (a) DSR-focused evaluation is a conceptual contribution addressing ISR's limitations; (b) target-free anonymization reduces systematic impersonation risks compared to targeted methods; (c) the reviewer's test protocol was incorrect (comparing protected image to its own original rather than to gallery image of same identity); and (d) under correct evaluation, MASQUE outperforms OPOM (92.0% vs. 46.0% identification, 86.9% vs. 23.8% verification). The disagreement centers on whether the reformulation of evaluation metrics and integration of existing components constitutes sufficient novelty for a top-tier venue.

**Reviewer Scores:**

**Current Scores:**
- **Reviewer `N3Wv`**: 4 (marginally below threshold) - likely to remain at 4, possibly increase to 5-6 if satisfied with runtime/fairness clarifications
- **Reviewer `JWTj`**: 4 (marginally below threshold) - likely to remain at 4, possibly increase to 5 if convinced by pairwise guidance justification
- **Reviewer `WCjs`**: 4 (marginally below threshold) - unlikely to increase without clearer articulation of core technical innovation
- **Reviewer `mg7j`**: 2 (reject) - unlikely to increase significantly given fundamental concerns about novelty and evaluation protocol disagreements

**Expected post-discussion scores**: 2, 4, 4-5, 4 (median: 4)

---

### Decision · Program_Chairs · 2026-01-26

Reject